# Memory-specific encoding activities of the ventral tegmental area dopamine and GABA neurons

**Vasileios Glykos[1,2]\*, Shigeyoshi Fujisawa[1]\***

[1]Laboratory for Systems Neurophysiology, RIKEN Center for Brain Science, Wako, Japan; [2]Synapse Biology Unit, Okinawa Institute of Science and Technology, Okinawa, Japan

**Abstract** Although the midbrain dopamine (DA) system plays a crucial role in higher cognitive functions, including updating and maintaining short-term memory, the encoding properties of the somatic spiking activity of ventral tegmental area (VTA) DA neurons for short-term memory computations have not yet been identified. Here, we probed and analyzed the activity of optogenetically identified DA and GABA neurons while mice engaged in short-term memory-dependent behavior in a T-maze task. Single-neuron analysis revealed that significant subpopulations of DA and GABA neurons responded differently between left and right trials in the memory delay. With a series of control behavioral tasks and regression analysis tools, we show that firing rate differences are linked to short-term memory-dependent decisions and cannot be explained by reward-related processes, motivated behavior, or motor-related activities. This evidence provides novel insights into the mnemonic encoding activities of midbrain DA and GABA neurons.

**\*For correspondence:**
vasileios.glykos@oist.jp (VG);
shigeyoshi.fujisawa@riken.jp (SF)

**Competing interest:** The authors declare that no competing interests exist.

## eLife assessment

This study characterized the activity of optogenetically identified dopaminergic and GABAergic neurons in the ventral tegmental area in mice performing a memory-guided T-maze task, and shows that subpopulations of dopaminergic and GABAergic neurons exhibited choice-related activity during the delay period, consistent with some previous studies (e.g. Morris et al., 2006, Parker et al., 2016). The authors demonstrate that these delay-period activities were enhanced when the task requires short-term memory. The results are **convincing** and this study provides **important** results regarding the nature of delay-period activity in the task.

## Introduction

Dopamine (DA) neurons originating in the ventral tegmental area (VTA) project to diverse forebrain regions, forming distinct but interacting neuromodulatory systems that are thought to play pivotal roles in the regulation of reward-related learning, motivation, and cognition (*Sawaguchi and Goldman-Rakic, 1991*; *Schultz et al., 1993*; *Goldman-Rakic, 1995*; *Schultz et al., 1997*; *Tzschentke, 2001*; *Schultz, 2002*; *Pierce and Kumaresan, 2006*; *Berridge, 2007*; *Vijayraghavan et al., 2007*; *Lammel et al., 2008*; *Robbins and Arnsten, 2009*; *Hauber, 2010*; *Cohen et al., 2012*; *Salamone and Correa, 2012*; *Howe et al., 2013*; *Matsumoto and Takada, 2013*; *Hamid et al., 2016*; *Mohebi et al., 2019*). A wealth of electrophysiological recordings from midbrain DA neurons, complemented by in vivo microdialysis data indicate that midbrain DA activity promotes behaviors associated with motivation (*Wise, 2004*; *Berridge, 2007*; *Salamone and Correa, 2012*; *Howe et al., 2013*; *Matsumoto and Takada, 2013*; *Hamid et al., 2016*; *Mohebi et al., 2019*) and supports reward-based learning

by encoding reward prediction error (RPE) signals (*Schultz et al., 1993*; *Schultz et al., 1997*; *Cohen et al., 2012*).

Also, DA is of central importance to higher cognitive functions, such as updating and maintaining short-term memory (*Sawaguchi and Goldman-Rakic, 1991*; *Miller and Cohen, 2001*; *Ott and Nieder, 2019*). Pioneering behavioral studies that pharmacologically manipulated the activity of DA receptors in the PFC revealed the significant role of DA signals on short-term memory. An inverted-U-shape effect was discovered, where too little or too much DA receptor stimulation impairs PFC-engaging short-term memory (*Sawaguchi and Goldman-Rakic, 1991*; *Vijayraghavan et al., 2007*; *Robbins and Arnsten, 2009*). Moreover, at the origin of the DA system, electrophysiological recordings at the VTA showed that DA neurons are not active in the delay period of memory tasks (*Schultz et al., 1993*; *Schultz, 2002*; *Phillips et al., 2004*; *Matsumoto and Takada, 2013*; *Choi et al., 2020*).

Motivated by the response of DA neurons to reward-related stimuli and memory delays, several lines of computational modeling studies sought to answer when and how DA signals support short-term memory 'update' and 'maintenance'. They proposed the 'gating theory', which provided a unified computational framework for reward prediction and short-term memory (*Cohen et al., 2002*; *Dreher and Burnod, 2002*; *Montague et al., 2004*; *Ott and Nieder, 2019*). According to the model, reward-predicting cues, elicit phasic DA release which opens the gate for the afferent signals to be stored in memory (update). But, in the delay period, low, tonic DA levels close the gate for interfering signals to enter the PFC and overwrite the short-term memory component (maintenance). Although the 'gating theory' fits adequately the behavior-unique responses of DA neurons to the coding schemes of short-term memory, it relies mainly upon empirical evidence of putative DA neurons and the longstanding consensus that short-term memory depends on the unbroken chain of persistent neuronal activity (*Durstewitz et al., 2000*; *Curtis and D'Esposito, 2003*).

However, recent advances in the study of the brain's functional organization suggest that persistent neuronal activity might not be the only candidate mechanism for the active maintenance of goal representation over short delays, leading to the proposal of new coding schemes for short-term memory (*Stokes, 2015*; *Miller et al., 2018*). One of these candidate mechanisms regards the memory-dependent dynamic changes in functional connectivity. Neural oscillations are abundant in the mammalian brain and are thought to offer the networking framework for the temporal organization of neuronal activity and information processing in short-term memory (*Uhlhaas and Singer, 2006*; *Buschman et al., 2012*; *Miller et al., 2018*). Calculating the phase coherence of neural oscillations between distributed brain regions, provides an estimation of the functional connectivity between them (*Fries, 2005*). Among other basal ganglia regions, the VTA engages dynamically in the large-scale network of brain systems that support memory-related information processing. Simultaneous electrophysiological recordings were performed in the PFC and the VTA while rodents executed memory-guided behavioral choices in a T-maze task (*Fujisawa and Buzsáki, 2011*). Neural oscillations (4 Hz) were prominent in both regions throughout the task, but their power and coherence were adaptively increased in memory delay. In a similar behavioral task, another short-term memory-related coding scheme was reported, this time at the single neuronal level. It was shown that while rodents navigate the maze, performing memory-guided decisions, PFC and parietal neurons differentiate their firing activities between opposite behavioral choices (*Fujisawa et al., 2008*; *Harvey et al., 2012*). To summarize, this novel empirical evidence from rodent studies on the T-maze behavioral apparatus complements the coding framework of short-term memory with more dynamic and adaptive information-processing mechanisms other than persistent activity.

In studying the role of DA neurons in short-term memory, we should take into consideration that the DA neuronal circuit is by no means self-contained and therefore it should not be investigated in isolation. Neurons utilizing GABA as a neurotransmitter constitute approximately 30% of the VTA neuronal population. The memory-related encoding properties of these inhibitory neurons have been largely overlooked, despite evidence of a strong inhibitory influence on neighboring DA neurons (*Nair-Roberts et al., 2008*; *Omelchenko and Sesack, 2009*; *Tan et al., 2012*; *van Zessen et al., 2012*) and well-established interconnections with the PFC circuit (*Carr and Sesack, 2000a*; *Carr and Sesack, 2000b*).

In light of the above, we wished to investigate with fine temporal and spatial resolution the firing activity of optogenetically identified DA and GABA neurons while mice performed a T-maze reward-seeking task with memory load. We took into consideration that (i) earlier studies analyzed either the

activity of putative DA neurons or drew inferences of the population activity from voltammetry and fiber photometry recordings (*Schultz et al., 1993*; *Schultz, 2002*; *Phillips et al., 2004*; *Matsumoto and Takada, 2013*; *Choi et al., 2020*), and (ii) field potentials (like the 4 Hz oscillations recorded in the VTA) stem mainly from phase-aligned excitatory or inhibitory post-synaptic potentials, whereas spiking activity is sparse (*Traub et al., 2004*; *Buzsáki, 2006*). (iii) A recent report revealed the causal relationship of DA activity with short-term memory, by inhibiting DA neurons with optogenetic tools. However, they did not report the encoding properties of single VTA DA neurons (*Choi et al., 2020*).

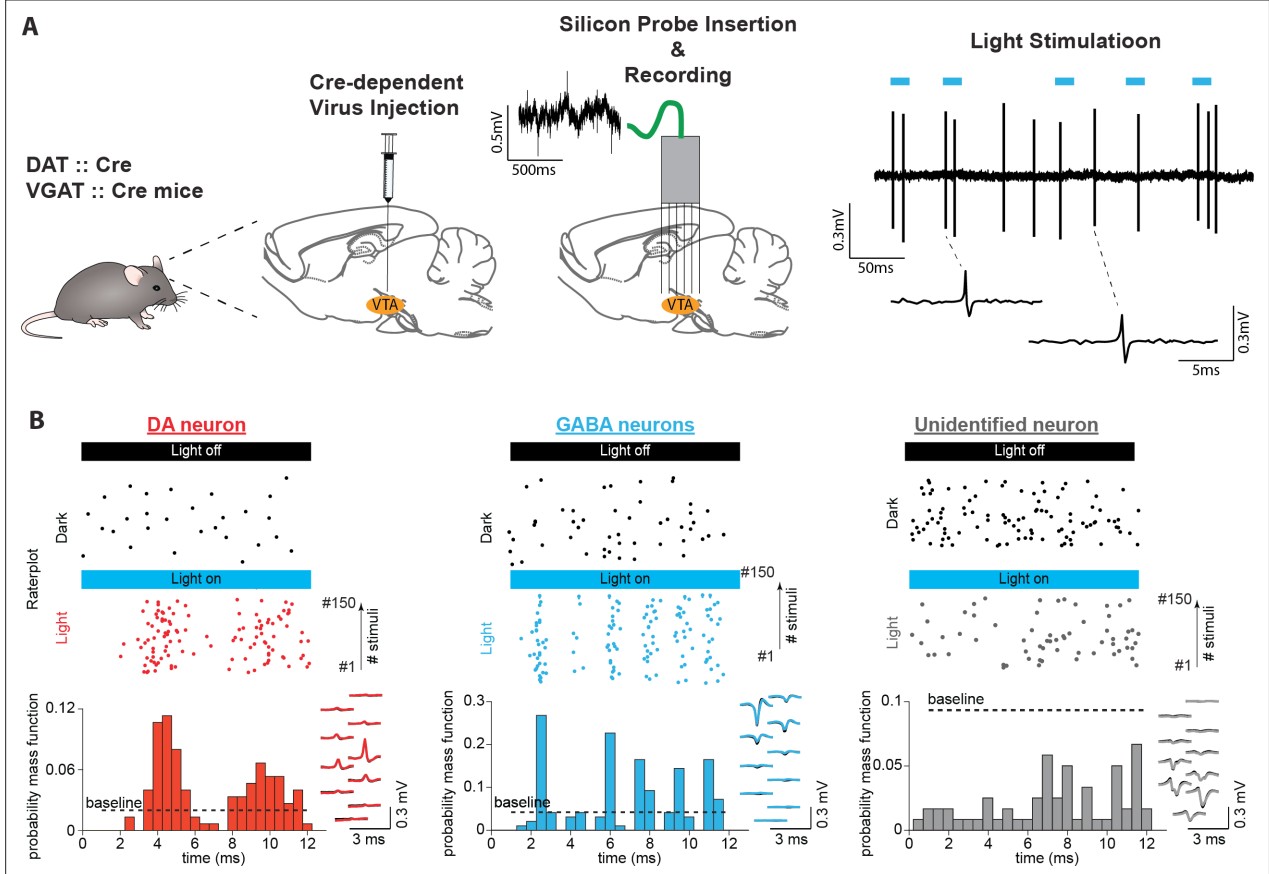

**Figure 1.** Identifying midbrain dopaminergic and GABAergic neurons. (**A**) Left: We confined ChR2 expression to DA and GABA neurons by injecting locally into the VTA the adeno-associated virus FLEX-ChR2 into transgenic mice expressing the Cre recombinase under the control of the promoter of the DA transporter (DAT::Cre) or the vesicular GABA transporter (VGAT::Cre). Approximately 10 days after the virus injection, the silicon probe was inserted into the brain in the same AP and ML coordinates. On a daily basis, the probe was inserted deeper into the brain by a few microns. Therefore, recording sessions were performed on different DV coordinates. Right: High-pass filtered voltage trace recorded during a light-stimulation session. Thick blue lines indicate light pulses (450 nm, 12ms). Two light-induced spikes are shown below. (**B**) Light response patterns of representative DA (red), GABA (blue), and unidentified (gray) neurons. (Top) Raster plots of spikes discharged during light stimulation (colored dots) and in the inter-stimulus baseline period (baseline, black dots). (Bottom) PSTHs extracted from the light-induced spikes. The black dashed line indicates the upper confidence limit of the baseline activity. If it is exceeded by the light-induced PSTH, then the unit is identified as light-responsive (See *Figure 1—figure supplement 2* for an explanation of this term). The right inset shows, superimposed, the mean waveforms of spontaneous (black) and light-induced (colored) spikes recorded by a single probe shank.

The online version of this article includes the following figure supplement(s) for figure 1:

**Figure supplement 1.** Validation of optogenetic identification results.

**Figure supplement 2.** Statistical method for optogenetic identification.

# Results

## Optogenetic identification of DA and GABA neurons in the VTA

In the present study, we sought to investigate the encoding properties of DA and GABA neurons of the VTA while mice engage in memory-dependent reward-seeking behavior. To identify neurons, we expressed the light-gated cation channel, channelrhodopsin-2 (ChR2), in DA and GABA neurons by injecting an adeno-associated virus containing FLEX-ChR2 into DAT-Cre and VGAT-Cre transgenic mice, respectively (*Bäckman et al., 2006*; *Tsai et al., 2009*; *Vong et al., 2011*; *Figure 1A*). Optogenetic identification and parallel electrophysiological recordings were performed using a custom-made diode-probe system diode-fiber assemblies attached to high-density silicon probes (*Stark et al., 2012*, *Figure 1A*, *Figure 1—figure supplement 1*). For each neuron, we assessed the response to light pulse trains delivered before and after behavioral sessions (*Figure 1*, *Figure 1— figure supplements 1–2*). We identified 104 neurons recorded from five DAT-Cre mice (hereafter referred to as DA neurons) and 74 neurons recorded from four VGAT-Cre mice (GABA neurons) with significant excitatory responses to light pulses (*Figure 1B*). Light-induced spikes from these neurons were almost identical to spontaneous spikes (waveform correlation coefficient >0.9, *Figure 1—figure supplement 1*). In addition, the electrophysiological profiles of the identified neuronal populations resembled those of previous studies (i.e. DA neurons fired action potentials with both wider wave-forms and slower spontaneous firing rates than GABA neurons; *Figure 1—figure supplement 1*), confirming the selective expression of ChR2 in DA and GABA neurons (*Cohen et al., 2012*; *Tan et al., 2012*).

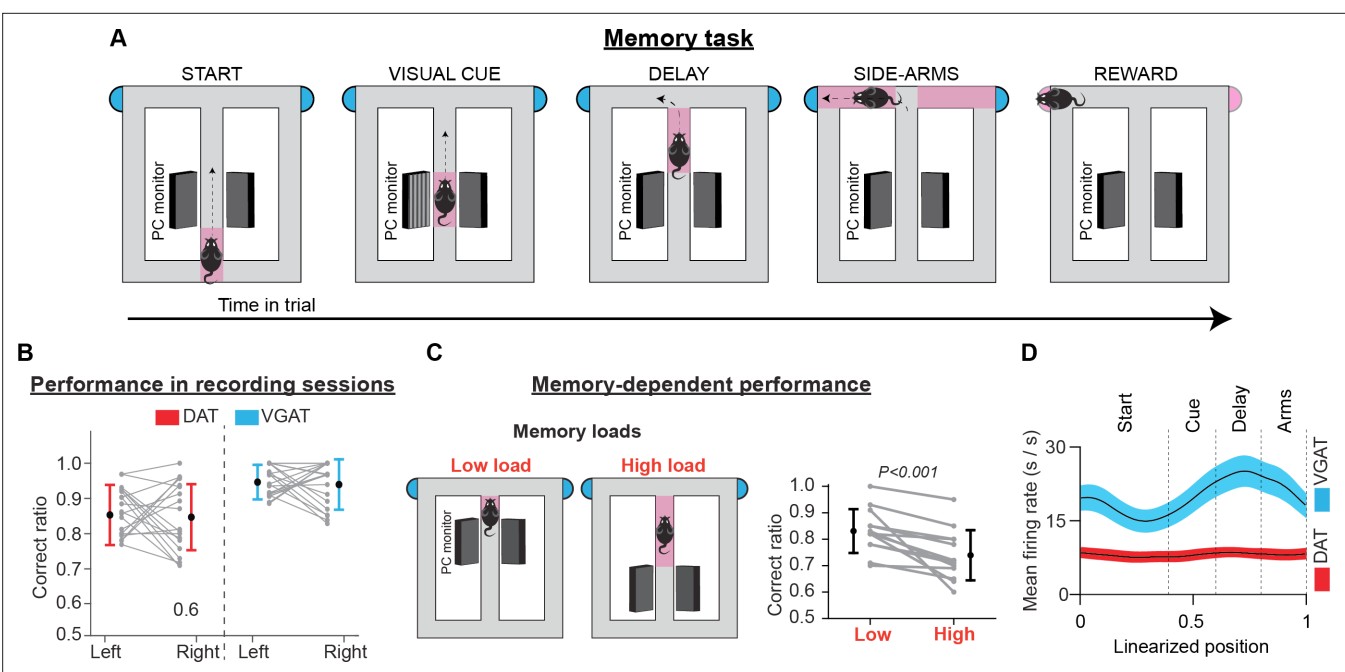

**Figure 2.** Memory task, performance, and population activity. (**A**) Schematic representation of the T-maze apparatus configuration in the memory task. Depending on the individual features of cognitive demand, the maze was divided into sections. Every trial commenced when the animal left the starting point, running along the central arm. In the visual cue section, a visual cue signaled the rewarded location. Between the visual cue section and the turning point at the end of the main arm, a brief memory delay was introduced. After reward consumption, the animals returned of their free will to the starting point to commence a new trial. (**B**) Correct performance rates in sessions with electrophysiological recordings. Gray lines illustrate performance rates for left and right trials in every session. Colored lines illustrate performance averages across sessions (mean ± standard deviation) for DAT-Cre (red) and VGAT-Cre (blue) animals. (**C**) In some training sessions animals received two blocks of trials with different memory loads. Gray lines illustrate correct performance rates for each block in every session and black lines show the average performance rates (mean ± standard deviation) across sessions, for all animals. (**D**) Mean firing rates (thick lines) ± 1 standard error of the mean (shaded areas) of the population activities of DA (red) and GABA (blue) neurons. The averaged population firing activity of GABA neurons increased in the cue and delay sections. However, the average population activity of DA neurons did not deviate from the beginning until the end of the trials.

The online version of this article includes the following figure supplement(s) for figure 2:

**Figure supplement 1.** T-maze configuration, behavior and running speed assessment in the memory task.

## Behavioral performance in a memory-dependent decision-making task

Mice were trained to perform sensory-guided and memory-dependent decisions in the "Memory Task" (*Figure 2A*, *Figure 2—figure supplement 1*). This task required animals to associate a visual cue presented at the beginning of the trial with a rewarded side arm of a figure-eight T-maze. A short memory delay was introduced between cue presentation and action selection. Following a correct response, they received water (5 µl) from a waterspout located at the end of each arm. Depending on the individual features of cognitive demand, the maze apparatus was divided into separate sections (i.e. 'start', 'cue', 'delay', 'side arms', and 'reward'). To ensure that the mice made choices guided by the visual cues and had minimal influence from other behavioral parameters on decisions, we eliminated imbalances between the left and right trials in key task parameters (e.g. reward amount, visual environment, effort, and motor skill requirements).

At the time of neurophysiological data collection, all mice performed memory task trials with high accuracy. Averaging across sessions, the total correct rate was 86.8 ± 7.9% (mean ± standard deviation [SD]; left: 88.1 ± 10.0%, right: 87.2 ± 11.6%; paired *t*-test evaluating left vs right performance rate: t(59) = 0.46, p=0.65, 60 sessions in nine mice, *Figure 2B*). In addition, performance was independent of individual preference for the left-or-right arm visits in any of the recorded sessions (test of independence, $\chi^2$(1)<3.84, p>0.05, Ho: correct rate is independent of arm choice, *Figure 2—figure supplement 1*).

We also assessed the contribution of memory-related processing to task performance. To achieve this, we delivered blocks of trials with different memory loads in separate training sessions. Across all sessions, the correct performance rate dropped with higher memory load demands (mean ± SD; low load: 83.1 ± 8.3%, high load: 73.9 ± 9.5%; paired *t*-test on correct performance rate: t(12) = 4.33, p<0.001, 13 sessions in seven mice, *Figure 2C*). This result is consistent with earlier reports (*Floresco and Phillips, 2001*; *Floresco and Magyar, 2006*) and highlights the important role of memory in supporting decisions in the present task.

## The population activity of DA neurons is not elevated during the memory task trials

DA neurons are not known to be active in the delay period of short-term memory tasks (*Schultz et al., 1993*; *Phillips et al., 2004*; *Matsumoto and Takada, 2013*; *Choi et al., 2020*), even though DA is a key neurotransmitter in the regulation of prefrontal cortical mnemonic functions (*Goldman-Rakic et al., 1989*; *Smiley et al., 1992*; *Smiley and Goldman-Rakic, 1993*; *Goldman-Rakic, 1997*; *Tzschentke, 2001*). This well-established notion has been established from either analysis of putative neuronal activities or inferred from voltammetry and fiber photometry recordings (*Ljungberg et al., 1991*; *Schultz, 2002*; *Phillips et al., 2004*; *Matsumoto and Takada, 2013*; *Choi et al., 2020*). Corroborating these earlier reports, the average discharge rate of identified DA neurons in the present study remained essentially constant (*Figure 2D*). Simple linear regression analysis, with the neuronal firing rate as the response variable and the animal's position on the maze as the single predictor variable, showed that from the beginning until the end of the trial (a 1.5-m distance), the population activity of DA neurons deviated slightly by 0.17±0.62 Hz (mean ± standard error of the mean [SEM], did not differ from a distribution with a mean equal to zero; one-sample *t*-test on the position coefficient, $t_{(103)}$ = 0.275, p=0.78). Notably, in the memory-delay period, the discharge rate of DA neurons declined by - 0.72±2.3 Hz (mean ± SEM, one-sample *t*-test on the position coefficient, $t_{(103)}$ = –0.31, p=0.75). On the other hand, the GABA neurons elevated their discharge rate by 4.29 ± 1.10 Hz in the delay period (mean ± SEM, one-sample *t*-test on the position coefficient, $t_{(73)}$ = 4.09, p<0.001), confirming evidence from an earlier report (*Cohen et al., 2012*).

## DA and GABA neurons in the VTA show trajectory-specific encoding preferences in short-term memory-dependent behavior

Making interpretations of the encoding properties of single neurons from population rate averages is highly challenging in tasks with many behavioral choices, especially for functionally heterogeneous populations such as the DA neurons. To overcome this limitation, we analyzed the firing activity of single neurons, by taking into consideration two important behavioral parameters. First, the animals visited either the left or right rewarded side arms in every trial. Therefore, we grouped and averaged trial spike trains of single neurons by the corresponding lap trajectories (left or right; see also

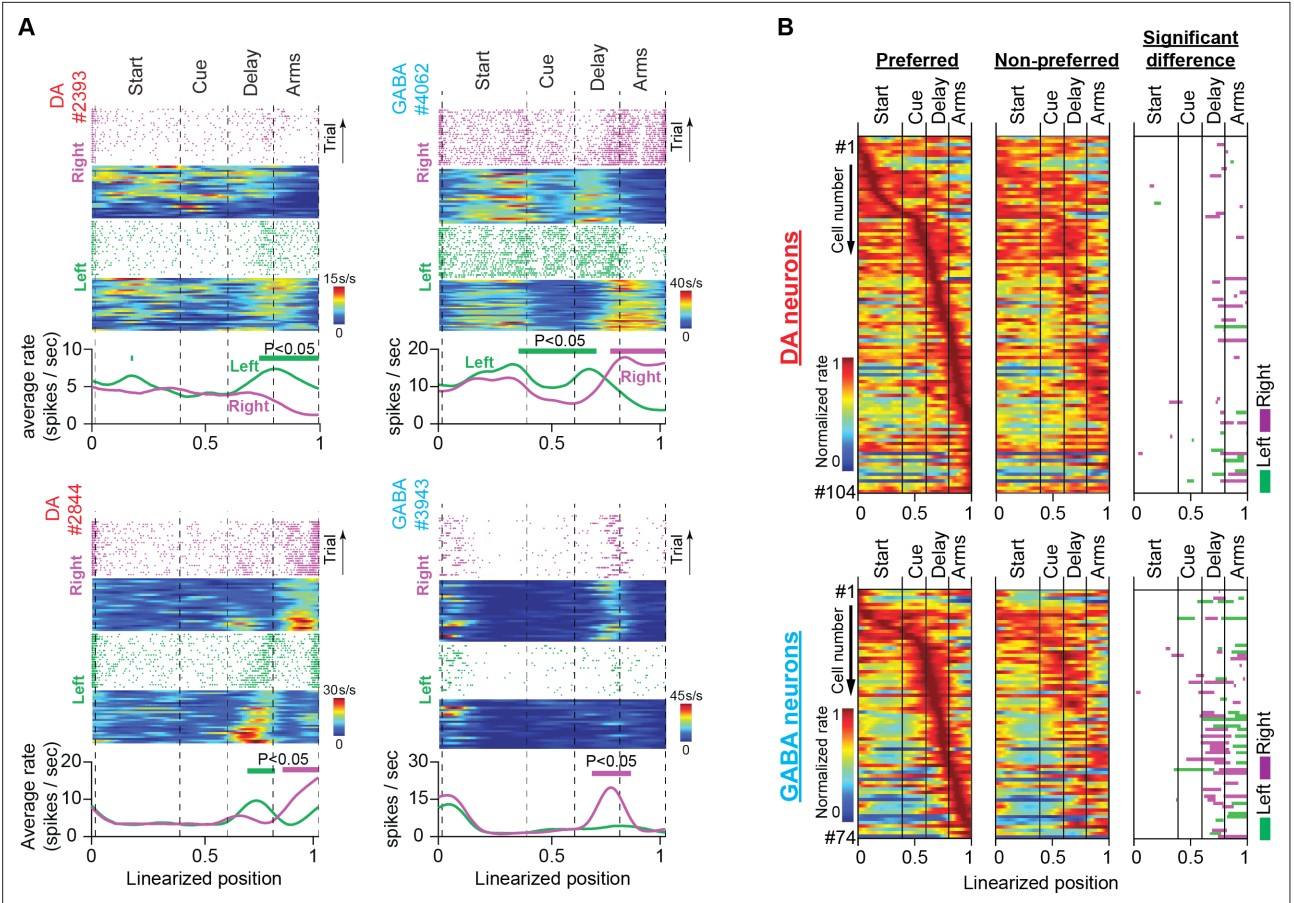

**Figure 3.** Trajectory-specific activities by DA and GABA neurons in the memory task. (**A**) Firing patterns of representative DA and GABA neurons. In each example: (Top) Raster plots of spiking events, for every correct trial, and their corresponding firing rate heatmaps as a function of position during right (purple) and left (green) trials. (Bottom) Average firing rates for correct left and right trials. Note that the trial and average firing rates (spikes/sec) are plotted as a function of position but normalized by the amount of time the mouse occupied each position on every trial. Thick lines above the average firing rates represent segments with significantly different firing rates between right and left correct trials (See also ***Figure 3—figure supplement 1***; permutation test; p<0.05). It is evident in these examples that midbrain neurons differentiate their discharge rates between left and right trajectories in certain positions. (**B**) Heatmaps of neuronal population responses organized by preferred lap trajectory (i.e. the trajectory with the stronger response; first column) and non-preferred lap trajectory (i.e. the trajectory with the weaker response; second column) for DA neurons (Top; n=104 units, 35 sessions in five mice) and GABA neurons (Bottom; n=74 units, 25 sessions in four mice). Each row contains preferred and non-preferred trajectory responses of the same neuron. In every row, both responses are normalized by the maximum rate of the preferred trajectory. The third column shows maze segments with significantly different discharge rates between preferred and non-preferred trajectories.

The online version of this article includes the following figure supplement(s) for figure 3:

**Figure supplement 1.** Statistical method for identifying trajectory-specific neurons (permutation method).

**Figure supplement 2.** Firing activities of midbrain DA and GABA neurons in the memory task.

**Figure supplement 3.** Evaluating DA and GABA neuronal responses to specific behavioral variables in the memory task through regression analysis.

**Figure supplement 4.** Arranging neuronal firing activities by time or position.

**Figure supplement 5.** Anatomical organization of memory specific VTA neurons.

Methods and ***Figure 3—figure supplement 1***). Also, in the present task, significant behavioral events (including visual cue presentation, memory delay, and reward delivery) were inherently bound to fixed positions in the maze (***Figure 2A***, ***Figure 3—figure supplement 1***). Thus, we arranged spiking events according to the position they occurred, to get an estimate of the behavioral correlates of neuronal activity. To this end, individual trial trajectories were linearized and represented as a one-dimensional vector consisting of 100 linearly spaced points (trial start: point 0; trial reward: point 100).

Examples of discharge patterns arranged by position and trajectory are shown in ***Figure 3A***, ***Figure 3—figure supplement 2***. These representative neurons elevate transiently their firing activity

**Table 1.** Number of neurons with trajectory-specific firing activities in the memory task grouped by maze section.
Data are presented for all recorded neurons and individually for optogenetically identified DA and GABA neurons; DA, dopamine; GABA, gamma-aminobutyric acid.

**Memory task: Significant difference in firing rate between left and right trials**

|  | start | cue | delay | side arms | reward |
|---|---|---|---|---|---|
| All (n=1191) | n=49 (4%) | n=78 (7%) | n=377 (32%) | n=505 (42%) | n=461 (39%) |
| DA (n=104) | n=4 (4%) | n=3 (3%) | n=22 (21%) | n=23 (22%) | n=24 (23%) |
| GABA (n=74) | n=4 (5%) | n=11 (15%) | n=35 (47%) | n=35 (47%) | n=34 (46%) |

at specific positions and do so consistently across trials. When we organized the normalized mean firing rates for the preferred and non-preferred lap trajectories of each neuron (i.e. trajectories with higher and lower firing rates, respectively) by the position of elevated transient activity (left and middle heatmaps in *Figure 3B*) we discovered that the position preference was uniform among neurons, producing a population sequential activity from the start until the end of trials. This result is in disagreement with several classical conditioning (*Schultz et al., 1993*; *Cohen et al., 2012*; *Tan et al., 2012*; *van Zessen et al., 2012*), instrumental learning (*Parker et al., 2016*) and delayed-response task studies (*Matsumoto and Takada, 2013*; *Choi et al., 2020*), which report homogeneous DA neuronal responses at key-task events, and in striking agreement with reports from the cortex (*Fujisawa et al., 2008*; *Harvey et al., 2012*).

However, in our opinion, the most important finding was that the representative neurons in *Figure 3A* and *Figure 3—figure supplement 2*, differentiated their responses between left and right trials at certain maze positions in a robust manner. To assess the trajectory-specific effects on neuronal firing activity, we used the permutation method (*Figure 3—figure supplement 1*). First, we calculated the original difference between the average firing rates in the left and right trials. We then randomly reassigned the trajectory labels (left or right) on the trial spike trains and produced the permuted average firing rate differences. If neurons were modulated by trajectory, the original and permuted firing rate differences were significantly different. Since spiking events were arranged by position, the permutation method could also detect positions with significant differences.

The right heatmap in *Figure 3B* summarizes the results from the permutation analysis applied to the populations of 104 DA and 74 GABA neurons. In both neuronal populations, there was abundant trajectory-specific activity, concentrated mostly in the delay and side-arm sections. Almost 20% of DA neurons differentiated their response between left or right trajectories in those maze sections (21% in the delay section and 22% in the side-arm section, 104 neurons, permutation test, $p<0.05$, *Figure 3B* and *Table 1*). In GABA neurons, the percentage was even higher, with almost 50% of these cells eliciting trajectory-specific activities (47% in the delay section and 47% in the side-arm section, 74 neurons, permutation test, $p<0.05$, *Figure 3B* and *Table 1*).

There have been reports of DA neurons discriminating between visual cues in a T-maze task (*Engelhard et al., 2019*) or choice selections in delayed-match-to-sample tasks (*Matsumoto and Takada, 2013*; *Choi et al., 2020*). However, in the present study, we did not detect different responses between left or right visual cues. Furthermore, neuronal preference for trajectories was not restricted to the turning point, which could indicate neuronal engagement in motor preparation for choice execution. Instead, it was spread in a wider area, covering a distance from the memory delay onset until the end of the side arms.

A plausible explanation for the trajectory-specific responses in the side arms is that neurons were under the control of the sensory, motor, or goal-directed behavioral processes triggered by the opposite trajectories (*Howe et al., 2013*; *Hamid et al., 2016*; *Mohebi et al., 2019*). However, in the memory-delay section, trajectories were identical for the left and right trials, which could be suggestive of the engagement of these neurons in short-term memory processing. Neuronal preferences to arm visits in memory delay are not uncommon in T-maze tasks. They have been reported in prefrontal and post-parietal cortical neurons and have been attributed to short-term memory-dependent decisions (*Fujisawa et al., 2008*; *Harvey et al., 2012*). So, is the trajectory-specific activity in our task reminiscent of internally generated, memory representations, or can be attributed to the well-known DA-linked neuronal computations (*Schultz, 2002*; *Cohen et al., 2012*; *Berke, 2018*; *Engelhard et al.,*

*2019*)? To test this hypothesis, we proceeded to a series of statistical analyses and control behavioral tasks.

## Multiple regression analysis confirms the trajectory-specific effect on DA and GABA neurons

We discovered that significant proportions of VTA neurons fired preferentially for left or right trajectories at specific locations on the maze when we arranged discharge patterns by arm visit and position. This result does not attest that trajectory and position alone contribute to the neuronal firing rate. Midbrain DA neurons are known to respond to a wide variety of behavioral parameters (i.e. choice accuracy, reward history, running speed, and distance to rewards *Engelhard et al., 2019*) which could also exert a significant effect on neuronal firing activity. However, their effect could be dampened due to the specific firing range arrangement.

Since these behavioral variables are difficult to control with behavioral tasks, we assessed their contribution to neuronal responses using multiple regression analysis (*Figure 3—figure supplement 3* and Appendix 1). We found that all the examined variables (lap trajectory, trial number, speed, trial accuracy, and reward history) contributed to the firing activities of neuronal subpopulations; however, only the lap trajectory predictor could explain better the trajectory-specific activities observed in the ~20% of DA and ~50% of GABA neurons that were identified with the permutation analysis.

## Memory-dependent but not motivated behavior is related to trajectory-specific activity in VTA neurons

Next, we investigated the contribution of short-term memory in decision-making on the trajectory-specific activity of VTA neurons. Memory-dependent decision-making depends on three major computational components. These are (i) sensory input gating, (ii) maintaining and manipulating memory contents and (iii) generating and executing appropriate motor plans (*Cohen et al., 2002*; *Dreher and Burnod, 2002*; *Montague et al., 2004*; *Ott and Nieder, 2019*).

We eliminated all three components in a variation of the memory task. Specifically, we trained mice in the no-cue-no-choice task, in which they were not presented with a visual cue and, therefore, could not make predictions about the location of the reward (*Figure 4A*). Furthermore, the choice selection was prevented by the presence of blocked side arms when they arrived at the T-intersection. After a short delay (approximately 1 s), access to one of the side arms (chosen pseudo-randomly) was permitted, which always led to a reward.

In recording sessions, mice received mixed protocols composed of randomly interleaved memory task and no-cue-no-choice task trials. We evaluated the trajectory-specific activities on each task separately using the permutation method. In the delay section of the no-cue-no-choice task, we observed a significant reduction in the number of positions with a significant firing rate difference (mean ± SD; DA: memory task 5.9±3.8 points, no-cue-no-choice task 1.5 ± 3.4 points, paired *t*-test, p=0.002, four animals; GABA: memory task 10.7 ± 3.9 points, no-cue-no-choice task 1.5 ± 2.0 points, paired *t*-test, p<0.001, three animals, *Figure 4B–E*). The attenuating effect on trajectory-specific activity was also reflected by a marked reduction in the number of trajectory-specific neurons (*Figure 4E*, numbers in parentheses). However, the firing rate difference between the left and right-side arms was strong in both tasks (DA: memory task 8.5 ± 7.8 points, no-cue-no-choice task 5.9±7.4 points, paired *t*-test, p=0.139, four animals; GABA: memory task 12.6±8.1 points, no-cue-no-choice task 9.9±8.8 points, paired *t*-test, p=0.234, three animals, *Figure 4B–E*).

However, the significant reduction in trajectory-specific encoding preference in the no-cue-no-choice task could not be entirely attributed to the absence of the memory component. This is because important running speed, motor responses, and motivational discrepancies exist between memory and no-cue-no-choice tasks. With regard to motivation, the important role of DA in adaptive decision-making is widely recognized (*Hamid et al., 2016*; *Berke, 2018*; *Mohebi et al., 2019*). We did not observe animal choice bias in memory task performance (*Figure 2B* and *Figure 2—figure supplement 1*), but we cannot rule out the possibility that individual neurons were modulated differently by effortful actions to reach the left- and right-sided rewards (*Figure 5—figure supplement 1*). Unlike the memory task, in the no-cue-no-choice task, the mice could not direct behavior towards the left- or right-side arms due to the absence of a visual cue. As a result, they were unable to allocate incentive motivational drives to the left-or-right trials (*Howe et al., 2013*; *Hamid et al., 2016*; *Berke, 2018*;

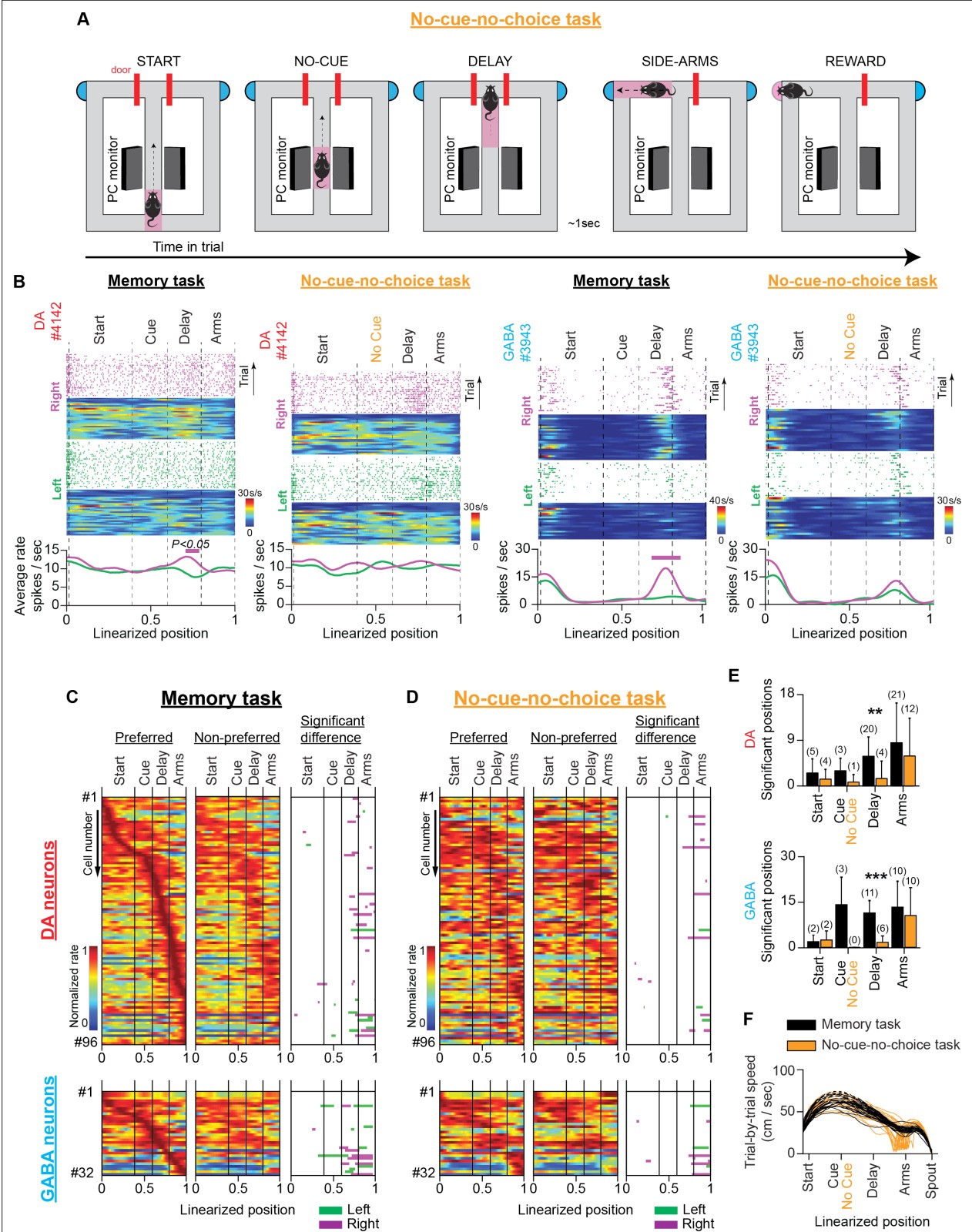

**Figure 4.** VTA neuronal responses in a T-maze task without visual cues and memory-dependent decisions (no-cue-no-choice task). (**A**) Schematic representation of the T-maze apparatus illustrating the sequence of events in the no-cue-no-choice task. (**B**) Firing patterns of representative DA (left) and GABA (right) neurons during the memory and no-cue-no-choice tasks. Both examples illustrate that the trajectory-specific firing rate difference in the delay section of the memory task declines prominently in the control task when animals do not receive visual cues that indicate the reward location,

*Figure 4 continued on next page*

*Figure 4 continued*

or enable memory-dependent decisions. (**C** and **D**) The firing patterns of DA neurons (**C**; n=96 units, 30 sessions in four mice) and GABA neurons (D; n=32 units, 12 sessions in three mice) in the memory task and the no-cue-no-choice task recorded in the same sessions. (Left and Middle columns) Normalized average neuronal responses for preferred (left) and non-preferred (middle) trajectories. The right column represents the maze segments with significantly different discharge rates. The row order of the neurons is the same for the memory task and the control task heatmaps. The data shown here for the memory task are a subset of those shown in *Figure 3B*. (**E**) Average number of position points (mean ± standard deviation) with a significant rate difference, arranged by maze section and behavioral task for DA (left) and GABA (right) neurons (** p<0.01, *** p<0.001, paired *t*-test comparing numbers of significant position points between tasks. Also, the numbers in parentheses describe the number of neurons with a significant rate difference). (**F**) Representative example showing the prominent difference in running speeds (cm/s) between the memory (black) and no-cue-no-choice (brown) task trials in a single session.

*Mohebi et al., 2019*). Also, the initial access denial to the side arms in the control task eliminates any potential differences in the motor preparation coding schemes (according to the 'gating theory') for the opposite arm visits in the memory task (*Engelhard et al., 2019*; *Ott and Nieder, 2019*). Finally, regarding speed, a representative example of running speed differences between the two tasks within a single session is shown in *Figure 4F*.

To dissociate the short-term memory component of neuronal activity from the modulatory effects of running speed, incentive motivation, and motor-related signaling, we trained mice in a second control task. The cue-no-choice task preserved the same running speed parameters (*Figure 5—figure supplement 2*), motor skill requirements, and physical effort demands (i.e. visual cues, maze shape, arm length, and reward amount were the same) as the memory task, but it prevented animals from making decisions. Accordingly, the animals were presented with the same visual cue as in the memory task, which indicated the side arm that was rewarded and enabled them to allocate incentive motivational drive to the left-or-right trials; however, they were always forced to visit the rewarded arm by blocked access to the unrewarded arm (*Figure 5A*). In the same recording session, the mice performed a separate block of memory task trials. Similar to the first control task, in the delay section of the cue-no-choice task we observed a significant reduction in the spatial extent of the firing rate difference (DA: memory task 5.4±3.6 points, cue-no-choice task: 0.4±1.1, paired *t*-test, p=0.011, four animals; GABA: memory task 10.7±6.3 points, cue-no-choice task 3.5±4.2 points, paired *t*-test, p<0.001, 1 animal, *Figure 5B–E*). In the side arms, however, the trajectory-specific effect remained strong and was not significantly different from the effect observed in the memory task (DA: memory task 8.1±10.3 points, cue-no-choice task 5.0±4.6 points, paired *t*-test, p=0.146, four animals; GABA: memory task 8.7±5.9 points, cue-no-choice 5.2±6.2 points, paired *t*-test, p=0.086, 1 animal, *Figure 5B–E*).

Together, these results suggest that trajectory-specific responses in the delay period of the memory task could reflect short-term memory representations linked to decision-making behavior and cannot be explained by running speed, motor, and motivation-related signaling differences.

## Neuronal activities in delay and reward are unrelated

DA neurons are known to be excited by rewards (*Schultz et al., 1993*; *Schultz et al., 1997*; *Cohen et al., 2012*; *Matsumoto and Takada, 2013*; *Engelhard et al., 2019*; *Choi et al., 2020*). In agreement with this notion, we discovered that 27 DA neurons (28% of 104 neurons, *Figure 6B* column 4; we defined the first second of reward consumption as the reward section.) responded to reward with significant excitation.

DA neurons are known to discriminate between rewards with different magnitudes and predictabilities (*Tobler et al., 2005*; *Morris et al., 2006*; *Matsumoto and Takada, 2013*). In the present study, the animals were offered equivalent options in terms of reward magnitude, uncertainty, and effort. Thus, we predicted the presence of a small number of reward-discriminating neurons. However, we found that 23% of DA neurons and 46% of GABA neurons differed significantly in their responses to left-or-right rewards (*Figure 6A and B* column 3, and *Table 1*; paired *t*-test comparing mean firing rates, p<0.05). This unexpected result raised the hypothesis that the trajectory-specific activities we observed in the memory delay were related to selective preference for rewards.

To test this theory, first, we correlated the average firing rate difference in the reward section with the average rate difference in the preceding maze sections. In both neuronal populations, we found a significant positive relationship between the reward and side-arm sections (*Figure 6C*; Pearson's

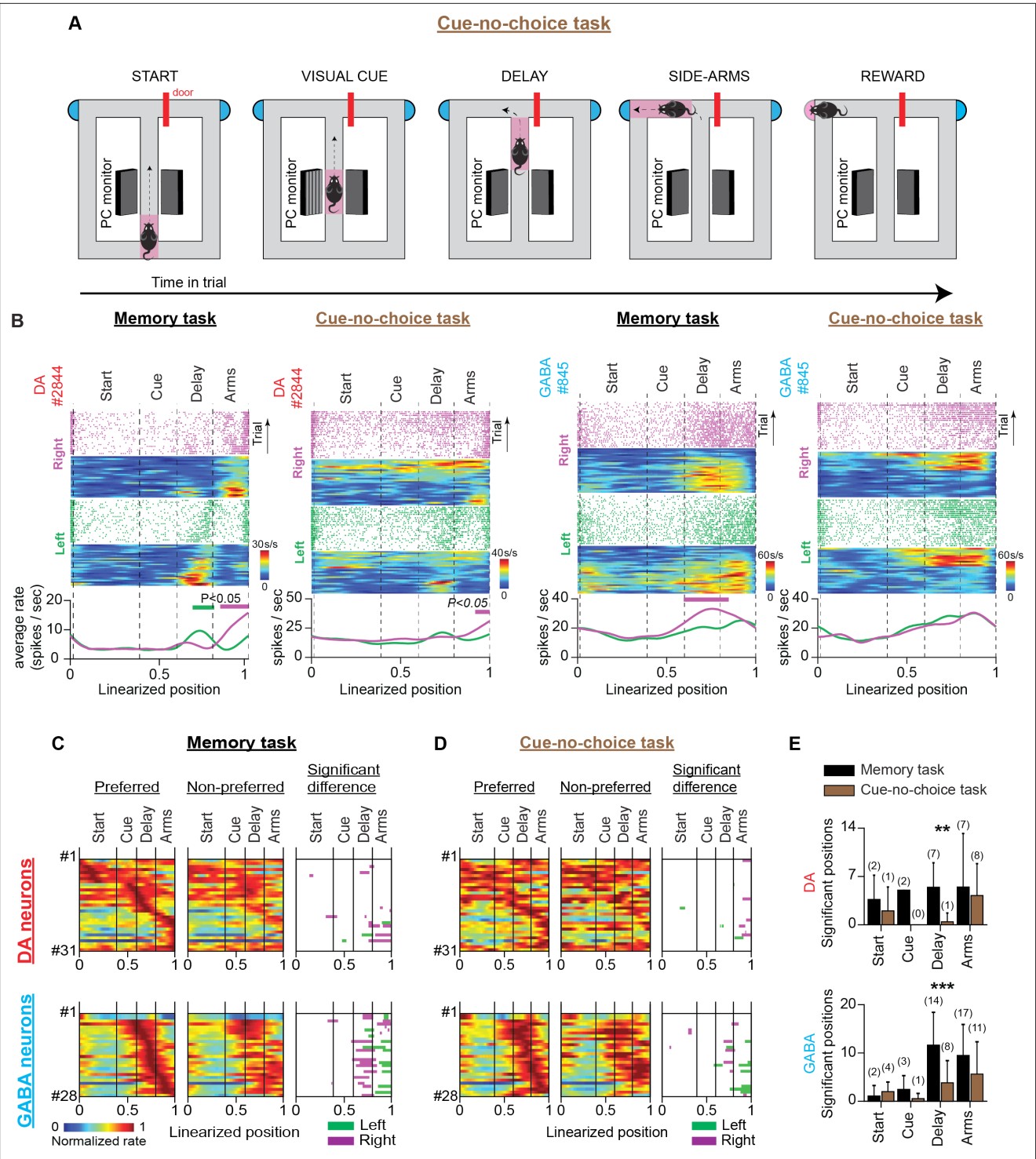

**Figure 5.** VTA neuronal responses in a T-maze task with visual cues but no memory-dependent decisions (cue-no-choice task). (**A**) Schematic representation of the T-maze apparatus illustrating the sequence of events in the cue-no-choice task. (**B**) Firing patterns of representative DA (left) and GABA (right) neurons during the memory and cue-no-choice tasks. Both examples illustrate that the trajectory-specific firing rate difference in the delay section of the memory task becomes notably weaker in the cue-no-choice task when animals do not make memory-dependent decisions, although running speed activities and incentive motivational drives of physical effort are the same between tasks. (**C** and **D**) The firing patterns of DA neurons (C; n=31 units, 28 sessions in four mice) and GABA neurons (D; n=28 units, 11 sessions in one mouse) in the memory task and the cue-no-choice task recorded during the same sessions. (Left and Middle columns) Normalized average neuronal firing rates associated with the preferred (left) and non-preferred (middle) trajectories. The right column represents the maze segments with significantly different discharge rates. The row order of the neurons

*Figure 5 continued on next page*

Figure 5 continued

is the same for the memory task and the control task heatmaps. The data shown here for the memory task are a subset of those shown in *Figure 3B*. (E) Average number of position points (mean ± standard deviation) with significant rate differences, arranged by maze section and behavioral task for (left) DA and (right) GABA neurons (** p<0.01, *** p<0.001, paired *t*-test comparing numbers of significant position points between tasks. Also, the numbers in parentheses describe the number of neurons with a significant rate difference).

The online version of this article includes the following figure supplement(s) for figure 5:

**Figure supplement 1.** The dopamine signaling model of incentive motivational drive.

**Figure supplement 2.** Running speed differences between the memory task and the cue-no-choice task.

correlation; DA: *R*=0.31, p<0.001; GABA: *R*=0.67, p<0.001) but not between the reward and delay sections (*Figure 6C*; Pearson's correlation; DA: *R*=–0.05, p>0.05; GABA: *R*=–0.03, p>0.05).

Next, we sought to determine whether neurons with trajectory-specific activities in memory delay, also exhibited a significant preference for the same-trajectory reward. To do so, we divided neuronal encoding preferences in six (6) categories determined by the maze section they elicited significant firing differences (delay or reward) and by the preference for trajectory (left-significant, right-significant, or non-significant). We discovered that only four (n=4 out of 104) DA neurons showed the same side preference in the reward and delay sections (e.g. elicit significantly stronger firing activities in the delay section of left trials and left reward). Thirty-one (n=31) neurons responded differently and sixty-nine (n=69) did not elicit significant responses in any of the sections (*Figure 6—figure supplement 1*).

The results from both analyses converge to the conclusion that trajectory-specific firing activities in memory delay do not reflect reward preference during consumption.

## Anatomical organization of trajectory-specific neurons

Several recent studies have reported that neighboring DA neurons are more likely to share similar encoding properties, thus, forming functional but also anatomical clusters within the VTA (*Lammel et al., 2008*; *Matsumoto and Takada, 2013*; *Engelhard et al., 2019*). Therefore, we sought to determine whether neurons with memory-specific encoding properties were anatomically segregated from the rest of the population. Estimating the location of recorded neurons (i.e. Bregma vs Mediolateral coordinates) from the anatomical reconstruction of the recording channels revealed that the trajectory-specific GABA neurons were located more laterally compared to the rest of the group (Bregma: $t_{(72)}$ = 1.165, p=0.248, Mediolateral: $t_{(72)}$ = –2.38, p=0.019, *Figure 3—figure supplement 5*). However, in the DA neurons there was no clear anatomical segregation (Bregma: $t_{(102)}$ = 0.045, p=0.964, Mediolateral: $t_{(102)}$ = 0.177, p=0.860, *Figure 3—figure supplement 5*). The lack of evidence of functional and anatomical segregation in the DA neurons could be accounted for by the fact that we targeted mostly the lateral parts of the VTA. Also, since the position of the recording channels along the dorsoventral axis was changing daily, we did not include in our analysis the dorsoventral coordinates of the recorded neurons.

## Discussion

In the present study, we performed extracellular recordings from optogenetically identified DA and GABA neurons in the VTA while mice performed reward-seeking tasks on a T-maze apparatus. Mice were trained to choose between two spatially separate goals under the instruction of visual cues presented at the beginning of the trial. A short memory delay was introduced between cue presentation and choice selection. We discovered that subpopulations of DA and GABA neurons showed differential responses between the left and right trials, starting from the onset of the memory delay in the main arm, where the trajectories were indistinguishable. Trajectory-specific preference was not correlated with reward history, running speed, the incentive motivational drive of physical effort, or reward-related encoding differences, and diminished significantly when the memory-dependent decision component was eliminated in control behavioral tasks. This evidence indicates that populations of DA and GABA neurons in the VTA encode internally generated signals that support short-term memory in decision-making.

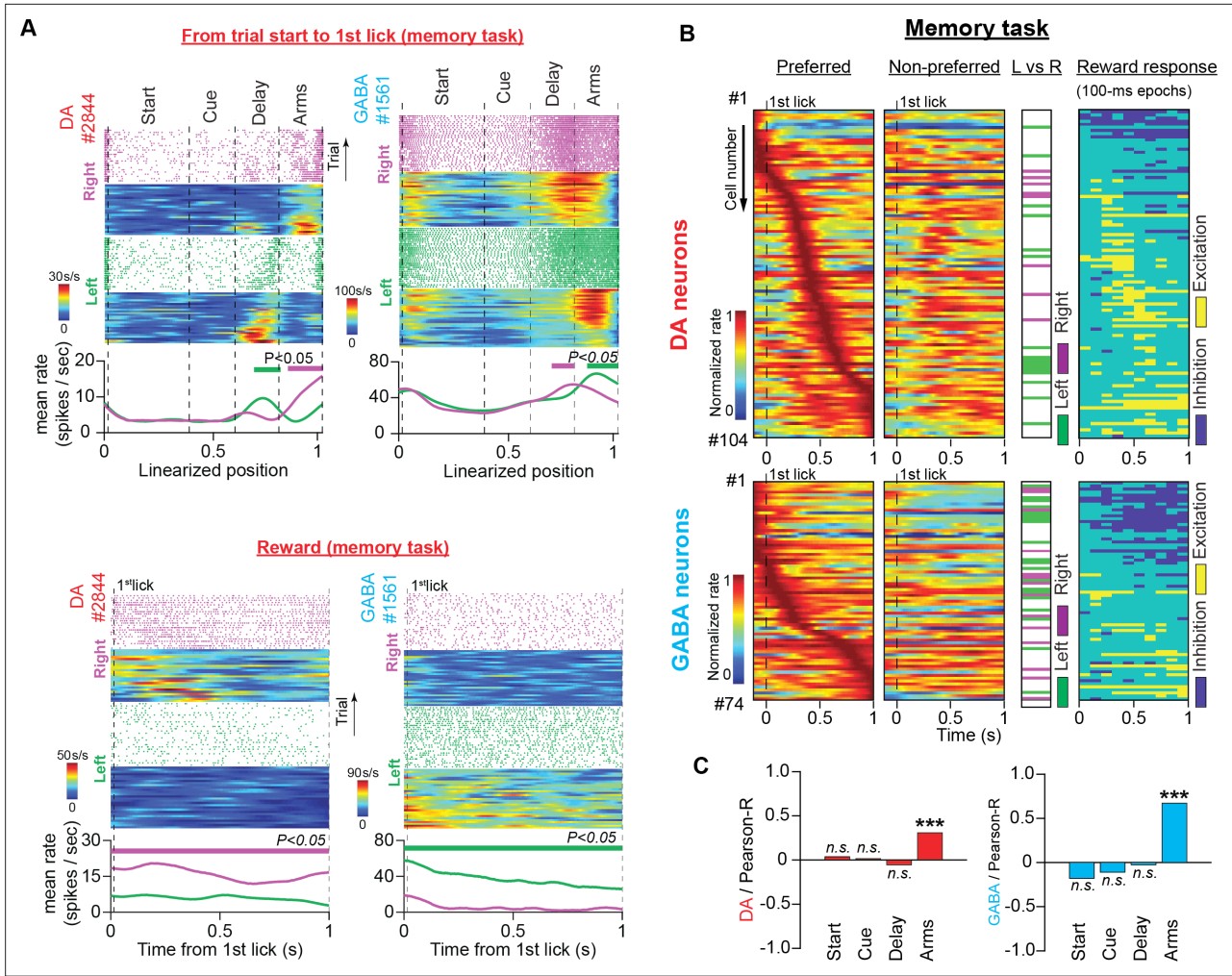

**Figure 6.** Neuronal responses during reward consumption are not related to the trajectory-specific activities in the memory delay. (**A**) Firing patterns of representative DA and GABA neurons in the memory task for the period from trial start until the first lick of the waterspout, which triggered the water pump (top, space domain) and during reward consumption (bottom, time domain). In each example: (Top) Raster plots of the spike trains and their corresponding firing rate heatmaps arranged by trajectory and position (maze) or time (reward) in right (purple) and left (green) trials. (Bottom) Average firing rates for correct left and right trials. The thick lines above the firing rates represent segments with significantly different firing rates between the correct right and left trials. (**B**) Firing patterns of DA (top) and GABA (bottom) neurons in the time domain during reward consumption (from the first lick until 1 s later) for preferred (first column) and non-preferred (second column) rewards (DA: n=104 units, 35 sessions in five mice; GABA: n=74 units, 25 sessions in four mice). Each row represents the normalized average firing rates (preferred and non-preferred) of a single neuron on a color scale. Neurons were ordered according to the time point of the maximum rate in the preferred arm. The third column shows neurons with significant discrepancies between the left and right reward-related responses (paired *t*-test for mean firing rates, *P*<0.05). The fourth column shows post-delivery reward segments (100ms each) with significant excitation or inhibition compared with the 100 ms pre-reward segment (paired *t*-test comparing firing rates, p<0.05). (**C**) Correlations between the mean firing rate difference in the reward section and the difference in every other maze section for DA (left) and GABA (right) neurons (Pearson's R values with p-values, *** p<0.001). Only the trajectory-specific firing rate difference in the side arms correlated with the reward-specific rate difference.

The online version of this article includes the following figure supplement(s) for figure 6:

**Figure supplement 1.** DA neurons which encode memory information for a specific trajectory do not show preference for the same trajectory reward.

## Activities of midbrain DA neurons in short-term memory

The 'gating theory' unifies the signaling activities of DA neurons in reward prediction and short-term memory (*Cohen et al., 2002*; *Dreher and Burnod, 2002*; *Montague et al., 2004*; *Ott and Nieder, 2019*). With regards to mnemonic processing, the well-established notion that DA somatic spiking activity is low in short-term memory stemmed either from recordings of putative DA neurons of the A8, A9, and A10 pathways (*Schultz et al., 1993*; *Schultz, 2002*; *Matsumoto and Takada, 2013*) or

inferred from neuronal population activities (*Phillips et al., 2004*; *Choi et al., 2020*). Consistent with the latter reports we did not observe a profound variation in the population activity of DA neurons during the memory task.

However, a wealth of recent studies has shown that DA neurons are functionally and genetically segregated (*Lammel et al., 2008*; *Lammel et al., 2011*; *Engelhard et al., 2019*). Moreover, in many real-life situations, animals must choose between many options for behavioral responses in high-dimensional environments. In such behavioral conditions, averaging neuronal responses, irrespective of the behavioral features and decisions they respond to, could hinder the fine computational processes of single neurons. To overcome this limitation, we analyzed the firing activities of identified single neurons, focusing on different discharge patterns between behavioral choices. Here, we demonstrated that memory-specific activities by midbrain DA neurons can be represented as trajectory-specific responses in the delay period of the memory task.

Other than gating sensory input, and maintaining and manipulating memory contents, DA has been implicated in relaying motor commands to elicit memory-guided responses (*Matsumoto and Takada, 2013*; *Engelhard et al., 2019*; *Ott and Nieder, 2019*). We tested this theory, but we found that the firing rate differences between left and right-arm responses declined in a control behavioral task (cue-no-choice) without memory load but with the same motor skill requirements as the memory task. From this, we concluded that motor preparation coding schemes for left or right arm responses (*Cohen et al., 2002*; *Ott and Nieder, 2019*) cannot account for the trajectory-specific activities in the memory delay of the T-maze task. Instead, our evidence indicates that trajectory-specific activities by DA (also GABA) neurons are functionally linked to the maintaining and manipulating of memory contents.

Our study provides more evidence to a mounting body of recent work that suggests a dynamic functional interaction between the PFC and VTA circuits in high-dimensional behavioral environments. The memory-related, trajectory-specific midbrain neuronal activities demonstrated here, have also been reported for PFC and post-parietal neurons while rodents perform reward-seeking responses on the T-maze (*Fujisawa et al., 2008*; *Harvey et al., 2012*). We also report that while mice navigate the maze, individual VTA neurons elevate transiently their firing activity at unique positions producing at the population level a neuronal sequential activity, which is a well-known physiological hallmark of cortical neurons. PFC and VTA networks are known to oscillate in high synchrony at 4 Hz in T-maze tasks (*Fujisawa and Buzsáki, 2011*). Network oscillations were also evident in our VTA recordings (unpublished data). Finally, we report that DA and GABA neurons of the VTA exhibit multitasking activities, encoding behaviors that overlap those of PFC neurons (see further discussion below). These striking similarities are in line with a recent computational theory proposing that the encoding properties of DA neurons reflect those of the upstream PFC neurons (*Lee et al., 2022*).

The present study also corroborates important findings from a recent report, which demonstrated that optogenetic perturbations in DA neuron excitability exert a strong effect on short-term memory performance, highlighting the causal role of DA neuronal firing activity in memory-dependent behavior (*Choi et al., 2020*). Also, in agreement with a previous report (*Engelhard et al., 2019*) by the same laboratory, we show that subpopulations of VTA neurons are modulated by running speed, cumulative performance rate, current choice accuracy, and reward history. Disparities between this and our study in the proportions of modulated neurons could be attributed to the different recording techniques applied as well as the maze regions of interest. Although, *Engelhard et al., 2019* trained mice in a virtual T-maze task, they analyzed neuronal firing activities and identified choice-specific neurons only in the visual-cue period, but not in memory delay. In contrast, we focused our analysis on the memory delay of the T-maze task.

Overall, our results agree with the notion that DA neurons encode a variety of behavioral parameters in complex environments. In addition, we confirmed that in memory-dependent behaviors, DA neuronal populations did not elicit sustained increases in their discharge rate. However, in the present task, DA neurons individually encoded internal representations by differentiating their responses to lap trajectories in memory delay.

## Role of motivated behavior in trajectory-specific encoding properties of VTA neurons

Midbrain DA activity is known to be involved in motivated behavior while rodents navigate mazes to receive rewards beyond immediate reach (*Hamid et al., 2016*; *Berke, 2018*; *Mohebi et al., 2019*). When mice approach rewards, striatal DA concentrations increase, scaling flexibly with reward size and proximity, which is proposed to reflect a neural correlate of a sustained motivational drive (*Howe et al., 2013*). To evaluate the role of motivated behavior in the trajectory-specific preference of midbrain neurons, we compared firing activities between a memory task and a control task without memory-dependent decisions (cue-no-choice task). Although in the cue-no-choice task, the behavioral parameters that determined the incentive motivational drives were the same as in the memory task (visual cues, maze shape, and reward amount), neuronal responses did not differ between the left and right trials. This result strongly indicates that incentive motivational drives (at least for physical effort) do not contribute to trajectory-specific activities of midbrain neurons during the delay period of the memory task.

## Memory-specific activities of the VTA neurons are not attributed to reward prediction error signaling

We also assessed the role of reward-related processing in the trajectory-specific activity of the midbrain neurons. When animals estimate the spatial proximity of distant rewards, DA neurons calculate RPE signals from state-value functions (*Hamid et al., 2016*; *Berke, 2018*; *Engelhard et al., 2019*; *Mohebi et al., 2019*; *Kim et al., 2020*). In the present study, the animals received ongoing visual input, facilitating the continuous estimation of reward proximity. Thus, DA neurons can potentially estimate RPE signals from successive state values assigned to each position on the maze track (*Figure 5—figure supplement 1*). Therefore, the difference in firing activity between the left and right trials could be the result of differences in the state-value functions assigned to these trajectories (*Hamid et al., 2016*; *Berke, 2018*). However, significant evidence contradicts this hypothesis.

First, the behavioral parameters that determine the state-value functions for the left and right trajectories were set to be identical in the cue-no-choice task and memory task, by preserving the same maze configurations and delivering equal amounts of reward. In addition, behavior in both tasks was cue-driven; therefore, animals could make predictions about the reward location and orchestrate behavior accordingly. However, we observed a prominent reduction in the firing rate difference between the left and right trials in the cue-no-choice task (*Figure 5*). Second, a significant subset of DA neurons (approximately 20%) responded differently to the left and right rewards in the memory task, although the same amount of reward was delivered. This unexpected finding raised the hypothesis that the encoding preference for reward could be reflected in the values of the preceding states in the maze and, therefore, could account for the trajectory-specific effect in memory delay. However, the differences in the firing activity elicited by the consumption of left or right rewards were unrelated to the firing rate difference in the delay section (*Figure 6C*). Also, we discovered a very small minority of neurons with the same trajectory preference in the memory delay and reward sections within the same trial (*Figure 6—figure supplement 1*). In conclusion, these findings indicate that the encoding preference for lap trajectories exhibited by midbrain DA and GABA neurons cannot be simply explained by discrepancies in RPE signaling.

## GABA neurons of the VTA and short-term memory

With the advent of highly selective identification and perturbation techniques, new evidence has emerged regarding the encoding properties and functional roles of local VTA inhibitory networks in reward processing and motivation. There are reports demonstrating that GABA neurons of the VTA suppress reward consummatory behavior (*van Zessen et al., 2012*), facilitate aversive behavior (*Matsumoto and Hikosaka, 2007*; *Tan et al., 2012*), and elicit sustained activities in the delay period between conditioned and unconditioned stimuli (*Cohen et al., 2012*). During these behaviors, the responses of DA and GABA neurons are often inverse, such that when GABA neurons are excited, neighboring DA neurons decrease their discharge rate. In particular, aversive stimuli excite GABA neurons, which then suppress the neighboring DA neurons (*Tan et al., 2012*). In addition, during reward consumption, GABA neurons are inactive (*Cohen et al., 2012*; *van Zessen et al., 2012*); however, when excited, they inhibit DA neurons and disturb consummatory behavior (*van Zessen*

*et al., 2012*). Finally, in classical conditioning tasks, DA neurons respond to rewards and reward-predicting stimuli, whereas GABA neurons remain silent during such events (*Cohen et al., 2012*). However, we demonstrated here that midbrain DA and GABA neurons elicit remarkably similar encoding properties. Both neuronal populations respond to short-term memory-specific activities manifested by encoding preferences for lap trajectories. Notably, though, GABA neurons are more strongly engaged in this dynamic encoding activity since almost twice as many inhibitory neurons responded differently to the left and right trials.

This result presents an activity paradox. Given the abundant and potent synaptic inhibition of DA neurons by neighboring GABA neurons (*Omelchenko and Sesack, 2009*), it was unexpected that both populations were highly active and similarly engaged in tasks. However, anatomical evidence provides a plausible explanation. Local inhibitory neurons form a dense network of local synaptic innervations that target the dendritic sites of DA and other GABA neurons (*Traub et al., 2004*; *Buzsáki, 2006*; *Omelchenko and Sesack, 2009*). Although potent and well-suited for coordinated network activity, this synaptic inhibition is not as strong as somatic inhibition (*Jhou et al., 2009*; *Omelchenko and Sesack, 2009*), and it has been suggested that it is not sufficient to suppress DA neurons when they receive a strong excitatory drive from extrinsic sources (*van Zessen et al., 2012*).

Finally, the stronger engagement of GABA neurons in trajectory-specific activity is an interesting observation that requires further investigation. In our opinion, future research on this topic should point to the direction of network oscillations. The VTA circuit is known to oscillate in memory-engaging behaviors producing frequencies of a wide spectrum; up to 100 Hz (*Fujisawa and Buzsáki, 2011*); also unpublished observations in our study. Although the mechanisms supporting circuit oscillations in the VTA are not well investigated, evidence from the prefrontal cortex (*van Aerde et al., 2008*; *Glykos et al., 2015*) and the hippocampal formation (*Traub et al., 2000*; *Mann and Paulsen, 2007*) demonstrate that the excitation of the local network of inhibitory neurons is crucial for the generation and maintenance of network oscillations.

## Conclusion

In summary, we optogenetically probed DA and GABA neurons in the VTA while mice performed a decision-making task with memory load. We discovered that both neuronal populations elicited memory-dependent preferences for left or right trajectories that could not be explained by motor activity, motivated behavior, or reward-related processes. This evidence indicates that VTA neurons encode mental representations to support short-term memory-dependent decisions and provides insights into novel sophisticated coding strategies employed by the midbrain DA and GABA neurons in reward-related behavior.

# Materials and methods

**Key resources table**

| Reagent type (species) or resource | Designation | Source or reference | Identifiers | Additional information |
|---|---|---|---|---|
| Strain, strain background (*Mus musculus*) | C57BL/6J-Slc6a3[tm1.1(cre)Bkmn]/J | The Jackson Laboratory | Jax #006660; RRID: IMSR_JAX:006660 | male |
| Strain, strain background (*Mus musculus*) | C57BL/6J- Slc32a1[tm2(cre)Lowl] | The Jackson Laboratory | Jax #0016962; RRID: IMSR_JAX:0016962 | male |
| Strain, strain background (AAV) | AAV5-EF1a-DIO-hChR2(H134R)-EYFP-WPRE-pA | UNC Vector Core | N/A | N/A |
| Antibody | Anti-Tyrosine Hydroxylase (rabbit polyclonal) | EMD Millipore | AB152 | 1/1000 |
| Antibody | Anti-GFP (mouse monoclonal) | Aves Labs | SKU: 75–131 | 1/500-1/1000 |
| Antibody | Alexa Fluor 546 IgG (Goat anti-chicken polyclonal) | Thermo Fisher Scientific | A-11040 | 1/500-1/1000 |
| Antibody | Alexa Fluor 488 IgG (Goat anti-mouse polyclonal) | Jackson Immunoresearch Laboratories, Inc. | AB_23338840 | 1/500-1/1000 |
| Chemical compound, drug | DAPI Fluoromount-G | SouthernBiotech | Cat No:0100–20 | N/A |

*Continued on next page*

*Continued*

| Reagent type (species) or resource | Designation | Source or reference | Identifiers | Additional information |
|---|---|---|---|---|
| Software, algorithm | Matlab | Mathworks | N/A | https://www.mathworks.com/ |
| Software, algorithm | KlustaKwik2 | *Kadir et al., 2014* | https://github.com/kwikteam/klustakwik2, copy archived at *Buccino, 2019* | N/A |
| Software, algorithm | RPvdsEX | Tucker-Davis Technologies | https://www.tdt.com/component/rpvdsex/ | N/A |
| Other | Silicon probe: 64-site, 6-shank | NeuroNexus | Buzsaki64spL | For electrophysiology recordings |
| Other | Optical fiber AFS50/125Y | Thorlabs, Inc. | https://www.thorlabs.co.jp/thorproduct.cfm?partnumber=AFS50/125Y | For light stimulation for optogenetic identification |
| Other | Blue laser diode | OSRAM Opto Semiconductors | PL450B | For light stimulation for optogenetic identification |
| Other | 256-channel Multiplexed Biosignal Amplifier | Amplipex, Ltd. | KJE-1001 | For electrophysiology recordings |
| Other | Figure-eight T-maze apparatus | O'Hara & Co., Ltd. | http://ohara-time.co.jp/ | For behavioral task |

## Contact for reagent and resource sharing

Further information and requests for resources and reagents should be directed to and will be fulfilled by the Lead Contact, Dr. Shigeyoshi Fujisawa (shigeyoshi.fujisawa@riken.jp).

## Animals

All experiments were approved by the RIKEN Institutional Animal Care and Use Committee. We used five adult male DAT-ires-Cre Jackson's Lab; stock #6660; *Bäckman et al., 2006* and four Vgat-ires-Cre Jackson's Lab; stock #16962; *Vong et al., 2011* mice backcrossed to C57BL/6 J. Animals were housed in separate cages on a 12 hr dark/light cycle and each performed the behavioral tasks at the same time of the day, between 11:00 and 17:00. In the cage, they were provided ad libitum food access but were restrained from water availability.

## Behavioral tasks

All behavioral tasks took place on a T-maze apparatus. More information about the maze configuration is provided in *Figure 2—figure supplement 1*.

### Memory task

Behavioral sessions commenced with the animal being placed at the 'starting position' (*Figure 2A* and *Figure 2—figure supplement 1*). Then, access to the main corridor was provided and the animal had to run through the 'start' section (0–50 cm) before it arrived at the maze segment surrounded by two PC monitors ('visual-cue' section, 50–80 cm). In this section, it was presented with a distinctive visual object (vertical black and gray bars) in one of the two monitors (the other monitor remained dark) indicates which side arm to visit to obtain the reward (i.e. left cue → left reward, right cue → right reward). In the third region of the central arm ('delay' section, 80–120 cm) both monitors turned dark. While running in the delay section, the animal had to maintain the reward-related information and based on that perform the action selection at the T-intersection. The intersection at the end of the main arm designated the end of the delay section and the beginning of the 'side-arms' section (120–150 cm) where the animal runs towards the reward position in anticipation of the reward. Reward (5 µl water) was delivered on correct trials at the end of the side-arms section from a waterspout. The first activation of the light-beam sensor at the waterspout triggered the water-delivery pump (Burkert, Ingelfingen, Germany), followed by reward consumption ('reward' section). After consuming the reward, the animal could return of its own will to the starting position, to commence a new trial.

Daily behavioral sessions consisted of 80–100 trials. Only animals with at least three successive sessions with an 80% performance ratio or more in the training phase were subjected to surgical operations.

## No-Cue-No-Choice task

Trials of this control task were delivered in recording sessions, interleaved with memory task trials. When the animal entered the visual cue section it was not instructed by the visual cue (*Figure 4A* and *Figure 2—figure supplement 1*). Also, access to both side arms was initially denied by closed sliding doors. Approximately 1 s after the animal arrived at the turning point, one of the sliding doors opened (pseudo-randomly) providing access to the reward. On every trial, the presentation or absence of the visual cue could instruct the animal about the task rules (i.e. memory task or no-cue-no-choice task).

## Cue-no-choice-task

The settings of this control task were the same as the memory task settings, except for the blockade of the unrewarded side arm (*Figure 5A* and *Figure 2—figure supplement 1*). Thus, the animals were always forced to perform correct choices. Because in both tasks, the same cue was presented, the animals could potentially be confused about the trial's task rules. Therefore, memory task and cue-no-choice task trials were delivered in separate sets within the same recording session. Accordingly, when the animals completed the set of cue-no-choice task trials (approximately 50 trials), they were automatically delivered with another set of memory task trials (approximately 50 trials).

Recording sessions lasted approximately 20–30 min.

## Intracranial surgeries and electrophysiological recording

The surgical process consisted of two separate operations. First, mice (DAT-ires-Cre or Vgat-ires-Cre) were surgically injected with 200–500 nl of adeno-associated virus AAV5-EF1a-DIO-hChR2(H134R)-EYFP-WPRE-pA University of North Carolina vector core facility; (*Tsai et al., 2009*) into the VTA stereo-taxically (from inferior cerebral vein AP:~6.65 mm, from midline ML: - 0.55 mm on the left hemisphere, from surface 4–4.5 mm, *Figure 1A*). Ten to fifteen days later, mice were implanted with a silicon probe in the same AP and ML coordinates (vertical insertion was intended, 0 degrees; *Figure 1A* and *Figure 1—figure supplement 1*). We used Buzsaki64spL (NeuroNexus, Ann Arbor, MI, USA) silicon probes which are composed of 6 shanks (10 mm long, 15 µm thick, 200 µm shank separation) and each shank has 10 recording sites (160 µm$^2$ each site 0.6–1.0 MΩ impedance). The silicon probe was attached to a custom-made micromanipulator and moved gradually to the desired depth position. On every probe shank, optic fibers were firmly attached to secure an accurate and firm insertion of the recording channels in the deep midbrain area (*Figure 1—figure supplement 1*). For experiments where light delivery was required, two of the optic fibers (shanks 2 and 5) were coupled with blue (450 nm) laser diodes (PL450B, OSRAM Opto Semiconductors). Light dispersion could potentially cover the axial and transverse span of all 64 channels (*Stark et al., 2012*).

During recording sessions, the wide-band neurophysiological signals were acquired continuously at 20 kHz on 256-channel Amplipex systems (KJE-1001, Amplipex Ltd, Hungary; *Berényi et al., 2014*). Following surgery, the probe was inserted 45 µm deeper into the brain daily, until it reached the VTA. Thereafter, the probe was moved deeper by 20 µm / day. The average recording coordinates for the DAT-Cre animals are 3.32±0.32 mm (mean ± standard deviation) rostrocaudal and 0.82±0.17 mm mediolateral, and for the VGAT-Cre animals, 3.52±0.29 mm rostrocaudal and 0.90±0.17 mm medio-lateral (*Figure 1—figure supplement 1*).

We cannot exclude the possibility that some neurons were recorded in successive sessions because clustering analysis was performed on individual sessions.

## Data analysis

Unless otherwise stated, data analysis was performed with custom-made programs designed in MATLAB with Signal-processing and Statistics toolboxes.

## Light-stimulation protocols for optogenetic identification

Light stimulation protocols were delivered before and after the behavioral tasks. They were composed of 1, 2, 3, and 4 mW blocks of 450 nm light pulses. Each block consisted of 150 square pulses (12ms

pulse duration; 0–1ms and 11–12ms contained artifacts) delivered at 1, 2, 3–10 Hz. Electrophysio-logical data recorded during light stimulation and behavioral protocols within a single session were merged and clustered together.

## Statistical analysis for detection of light-responsive units

Neurons with light-induced responses exceeding the average spontaneous activity were classed as light-responsive. To identify light-responsive neurons we applied the statistical analysis described in detail in *Figure 1—figure supplement 2*.

## Estimation of firing activity during behavior

To estimate the neuronal firing activity while animals performed the behavioral task, we took into consideration the primary goal of this study; which is to look for trajectory-specific encoding prop-erties, as well as the inherent limitation of the task; that is the experimenter could not control the temporal precision of the behavioral events. To overcome this limitation, we arranged firing activity by the animal's position on the maze. To do so, first, we linearized the trial trajectories and assigned them with a lap trajectory label (left or right). Then, the linearized products were divided into 100 position points and normalized so that position 0 corresponded to the starting point of the trial and position 1 to the waterspout. Second, we constructed post-distance histograms, analogous to the peri-stimulus-time-histograms (PSTHs), although the time of spiking events was replaced by the posi-tion they occurred (for simplicity, also by habit, we will call the post-distance histograms as PSTHs). To construct accurate PSTHs we considered the exact position the spikes were discharged and the time the animal occupied this certain position. Let $n^{(k)}(x)$ be the number of spikes of a single neuron and $t^{(k)}(x)$ be the occupation time in the $x_{th}$ position point of the $k_{th}$ trial (*Figure 3—figure supplement 1*). Then, $\hat{\lambda}(x) = \frac{1}{K} * \sum_{k=1}^{K} \left[ n^{(k)}(x) / t^{(k)}(x) \right]$ where $K$ is the number of trials, represents the average firing rate probability (spikes / sec) at position point $x$. To examine the trajectory-specific encoding properties of VTA neurons we produced average firing rate histograms for correct left and right trials, separately. Then, both histograms were smoothed with a Gaussian Kernel function (σ=0.5, length of 20 position points).

## Firing rate heatmap construction

To construct the normalized firing rate heatmaps shown in *Figures 3B, 4C, D, 5C, D and 6B*, *Figure 3—figure supplements 2 and 4* we took the following steps. First, for every neuron we produced the average firing rate for left and right correct trials. Second, we normalized both rates by dividing them with the maximum firing rate of the strongest trajectory response (e.g. for the example shown in *Figure 3—figure supplement 1* we divided both average firing rates by the maximum rate of the response to the left trials). Then, the normalized rate of the stronger trajectory response was assigned to the 'preferred' heatmap and the rate of the weaker trajectory response to the 'non-preferred' heatmap (e.g. for the example shown in *Figure 3—figure supplement 1*, the left normalized rate was assigned to the 'preferred' heatmap and the right rate to the 'non-preferred' heatmap). Both rates occupied the same row. The row ordering was determined by the position of maximum rate.

## Identifying trajectory-specific neurons with the permutation method

To identify neurons with trajectory-specific encoding properties we applied the permutation test reported elsewhere (*Fujisawa et al., 2008*) and described in detail in *Figure 3—figure supplement 1*.

## Regression analysis

We designed a generalized linear regression model (GLM) with the neuronal firing rate (FR) modelled as a gaussian function of the lap trajectory (T), speed (S), trial number (TN), performance (R), current trial accuracy ($A_0$), and previous trial accuracy ($A_{-1}$) behavioural variables. With the permutation anal-ysis, we observed that the trajectory-specific effect on the firing activity was dependent on position. Thus, we examined the joint effect of trajectory with position (P) on spiking activity. All dependent and independent variables were arranged by position. The values of the trajectory (1 for left and 2 for right), trial number, performance (cumulative correct rate), current trial and previous trial accuracy

(1 for correct trial, 0 for error trial) variables remained constant throughout the whole trial. The firing rate, position and speed variables changed their values on every position.

The GLM was:

$$FR = \beta_0 + \sum_{k=1}^{6} \beta_{T*P} T \cdot P^k + \sum_{k=1}^{6} \beta_S \cdot S^k + \beta_{TN} \cdot TN + \beta_R \cdot R + \beta_{A_0} \cdot A_0 + \beta_{A_{-1}} \cdot A_{-1} + \varepsilon$$

where the β values are the regressor coefficients for the different predictors (including the intercept $\beta_0$) and ε is the Gaussian noise term. The 6th degree order polynomials of position and speed were chosen for model optimization with the Bayes information criterion.

First, we generated model predictions of the average firing rates for left ($L_0$) and right ($R_0$) trials, and from those we calculated the predicted firing rate difference ($D_0$). Then, we shuffled the trajectory labels assigned to the tested variable (the assigned labels to the rest of the independent variables remained intact) and assessed the effect on the firing rate difference. For every predictor we produced 500 shuffled rate differences, $D_j$. If the absolute mean value of $D_0$ exceeded the top 5% of the $D_j$ values (including Bonferroni correction), then the hypothesis was rejected, and the predictor was significantly contributing to the firing rate difference. We examined every maze region individually, but here we report only for the delay region.

## Reward-related excitation or inhibition

The reward section was defined as the first second of reward consumption. To assess neuronal response to reward consumption and categorize it as excitatory, inhibitory, or non-responsive we performed the following analysis. First, we produced the smoothed mean firing rate response in the time domain (as we did in the maze sections in the space domain) for left, $\lambda_{Left}(t)$, and right, $\lambda_{Right}(t)$ trials. For the preferred arm of each neuron, we compared the mean firing rate in the reward section to the mean rate in the 100ms epoch preceding reward delivery (paired t-test on mean firing rates; p<0.05; *Figure 6B* column 4).

## Encoding preferences in the reward section

The difference in the intensity of neuronal firing activity between left and right rewards was assessed by comparing the mean firing rate of neuronal activity elicited in the reward section of left and right trials (paired t-test on mean firing rates; p<0.05; *Figure 6B* column 3).

## Relationship of encoding preferences in the reward section to those in the remainder of the maze

To assess whether the trajectory-specific firing activity in the maze was linked to discrepancies in the response to left and right rewards, we followed the next steps of analysis. First, for every neuron and every maze section, we calculated the mean value of the relative firing rate difference between left and right trials ($D_{start}(x)$, $D_{cue}(x)$, $D_{delay}(x)$, $D_{arms}(x)$, $D_{reward}(t)$). Then, for each neuronal group, we calculated the linear relationship (Pearson's correlation) between the reward section mean values, to those in the remainder of the maze (Pearson's correlation; *Figure 6C*).

## Immunohistochemistry

After completion of the recording sessions, which lasted about a month, mice were anesthetized with isoflurane and perfused transcardially with 10 ml PBS and 10 ml paraformaldehyde (4%), before they were decapitated. Brains were then removed, post-fixed and coronal slices (100 μm) were prepared. The primary antibodies used were rabbit anti-tyrosine hydroxylase (TH) and chicken anti-GFP. The secondary antibodies used were AlexaFluor 549 anti-rabbit and 488 IgG anti-chicken, respectively. Sections were further stained with DAPI to visualize nuclei. Image acquisition was performed with a fluorescence microscope NanoZoomer (Hamamatsu, Japan) system.

## Methodological considerations

### Arranging firing rate by position

With only a handful of exceptions (our report belongs to this minority group), scientific manuscripts reporting the encoding properties of DA neurons arrange neuronal responses by time and align them by key behavioral events, such as trial start, visual cue presentation, reward delivery, etc. We also attempted to arrange firing activities by time, but soon we came to the realization of the inherent caveats of this method in the T-maze task.

The encoding properties of DA neurons have mostly been studied in classical conditioning or reinforcement learning tasks. In those tasks, the animal does not control the exact timing of behavioral events. The experimenter determines when trials begin or end, when visual cues or rewards are delivered, and how much time elapses between cue onset and rewards. Then, usually, post-hoc analysis looks for neuronal responses that are time-aligned to significant task events. For this type of analysis, the researcher should be careful so that the epochs of interest between successive events must not overlap, and most of all, important events are not captured within the epoch of interest of the preceding or succeeding events.

However, in the T-maze task, the timing of behavioral events was completely controlled by the animal, causing a high amount of variability in the timestamps of the task events. For example, within a recording session, animals would consume unique trial rewards within a range of 1–10 s. Also, the trial duration would usually increase throughout the session as the animals progressively became tired and slower. On top of all, there was high behavioral response variability between animals. Therefore, the increased variability in the timing of events between trials, sessions, and animals caused a significant amount of ambiguity when firing activities were arranged by time and aligned by key task events. *Figure 3—figure supplement 4* shows the average firing activity of individual neurons extracted from a single recording session when neuronal firing was arranged by time (A) and by position (B; the position-arranged firing rate of these particular neurons are presented in *Figure 3A*, but for convenience to the reader are also shown here). In *Figure 3—figure supplement 4* firing activities for left/right trials were also aligned at the timing of the delay offset (sensor 5). The shadow-colored areas show the range of the timestamps of the other important task events (trial start, cue onset, etc.) across 80–100 trials, relevant to the delay offset. It is evident from this example that deciding the epoch of interest for the memory delay is highly ambiguous.

This important caveat could be easily resolved by arranging firing activities by position. This way, we could produce reliable neuronal firing averages from recording sessions and perform comparisons between behavioral tasks and animals.

### Identifying and characterizing RPE signals in the Tmaze task

The role of DA in processing RPE signals has been studied extensively with classical conditioning tasks (*Ljungberg et al., 1991*; *Schultz et al., 1993*; *Schultz et al., 1997*; *Schultz, 2002*; *Tobler et al., 2005*; *Kim et al., 2020*). In this Pavlovian paradigm, animals are usually physically restrained and are not trained to make decisions, also, they are exposed to a strictly controlled sensory environment and receive easily accessible and immediate rewards. The present study was designed to investigate the memory-encoding properties of individual neurons in a high-dimensional environment. We did not observe strong manifestations of RPE signaling (*Figure 3—figure supplement 4*). However, compared to classical conditioning studies, in the T-maze task neurons were under the control of numerous behavioral parameters that could be masking cue-related responses, and therefore we cannot draw safe conclusions about the computational role of DA neurons on RPE signals. So far, only manipulating the behavioral parameters with virtual reality tools, has provided insight into the RPE-related responses of DA neurons in goal-directed behavioral tasks like ours (*Kim et al., 2020*).

## Acknowledgements

This study was supported by the Ministry of Education, Culture, Sports, Science, and Technology (Grants-in-Aid for Scientific Research 18H02711 and 18H05525), the Mitsubishi Foundation, the Naito Foundation, and the Japan Agency for Medical Research and Development (AMED).

In memory of Miles Adrian Whittington. A true mentor and friend.

# Additional information

## Funding

| Funder | Grant reference number | Author |
|---|---|---|
| Japan Society for the Promotion of Science | 18H02711 | Shigeyoshi Fujisawa |
| Mitsubishi Foundation | | Shigeyoshi Fujisawa |
| Naito Foundation | | Shigeyoshi Fujisawa |
| Japan Agency for Medical Research and Development | | Shigeyoshi Fujisawa |
| Japan Society for the Promotion of Science | 18H05525 | Shigeyoshi Fujisawa |

The funders had no role in study design, data collection and interpretation, or the decision to submit the work for publication.

## Author contributions

Vasileios Glykos, Conceptualization, Data curation, Software, Formal analysis, Investigation, Methodology, Writing - original draft, Project administration, Writing - review and editing; Shigeyoshi Fujisawa, Conceptualization, Supervision, Funding acquisition, Investigation, Writing - original draft, Project administration, Writing - review and editing

## Author ORCIDs

Vasileios Glykos ![ORCID] http://orcid.org/0000-0003-3382-5116
Shigeyoshi Fujisawa ![ORCID] http://orcid.org/0000-0003-3963-4281

## Ethics

All experiments were approved by the RIKEN Institutional Animal Care and Use Committee (Approval number: W2022-2-011).

Reviewer #1 (Public Review): https://doi.org/10.7554/eLife.89743.3.sa1
Reviewer #2 (Public Review): https://doi.org/10.7554/eLife.89743.3.sa2
Author Response https://doi.org/10.7554/eLife.89743.3.sa3

---

# Additional files

## Supplementary files

• MDAR checklist

## Data availability

Data files and analysis scripts included in this manuscript are at the GitHub repository (https://github.com/vglykos/Memory-specific-encoding-activities-of-the-ventral-tegmental-area-dopamine-and-GABA-neurons/tree/main copy archived at *Glykos, 2024*).

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

## Appendix 1

### Evaluating the contribution of behavioral variables in trajectory-specific activity with multiple regression analysis

We designed a generalized linear regression model in which the dependent variable was the position-arranged firing rate, and the independent variables (predictors) were the animal's running speed, trial number, performance rate (cumulative correct rate), current trial accuracy (reward or not), previous trial accuracy (reward history), and lap trajectory (left or right). Since the permutation analysis of the original spike trains revealed that the trajectory-specific effect on neuronal firing activity was highly dependent on the animal's position (occurring mainly in the delay and side-arm sections), we included the joint effect of lap trajectory and position in the training model instead of testing the effect of trajectory alone. A major advantage of regression analysis is that one can dissociate the inherently bound effects of the independent variables. For example, we can examine the influence of the lap trajectory variable on neuronal firing activity without including the effect of the running speed variable (which could potentially differ between left and right trials).

For each neuron, we produced model predictions for the correct left and right trial average firing rates (*Figure 3—figure supplement 3*, dashed lines) from which we calculated the predicted firing rate difference. We subsequently examined the contribution of each independent variable to the trajectory-specific firing activity by shuffling the trajectory labels assigned only to this particular variable. If the tested variable exerts a significant effect on the firing rate, then shuffling the trajectory labels would produce a significant reduction in the predicted firing rate difference. We examined the same pool of neurons that we reported on the memory task, as shown in *Figure 3B* (n=104 DA and n=74 GABA-identified neurons). In the delay region, the joint effect of lap trajectory and position (trajectory ×position predictor) contributed significantly to the predicted average rate difference of 23 DA neurons (*Figure 3—figure supplement 3B*, C top), 17 of which overlapped with the 22 trajectory-specific neurons identified with the permutation analysis (*Figure 3—figure supplement 3C* bottom). Of the GABA neurons, 34 were modulated by trajectory and position (*Figure 3—figure supplement 3B*, C top), 30 of which were also trajectory-specific (*Figure 3—figure supplement 3C* bottom). These results confirm that the trajectory factor was responsible for the firing rate difference shown in *Figure 3B*. The speed variable significantly modulated 24 DA and 22 GABA neurons (*Figure 3—figure supplement 3B*, C top), but only four DA and four GABA neurons were co-modulated by trajectory (*Figure 3—figure supplement 3C* bottom). The performance and accuracy variables modulated smaller numbers of neurons (*Figure 3—figure supplement 3B*, C top), and the reward outcome of the previous trial did not co-modulate any of the trajectory-specific DA and GABA neurons (*Figure 3—figure supplement 3C* bottom). The trial variable modulated 30 DA neurons and 32 GABA neurons, co-modulating with the trajectory variable of 8 DA and 15 GABA neurons. However, the distribution of the trial predictor coefficient did not differ from a distribution with a mean equal to zero (one-sample $t$-test, $P>0.05$, *Figure 3—figure supplement 3D*), indicating that the effect of successive trials on firing rate did not reflect cognitive processing, but was caused by mechanical reasons; due to the animal's movements, the distance of the recording channel from the targeted neurons changed continuously, which affected the signal-to-noise ratio and eventually spike detection. In agreement with Engelhard et al., a notable proportion of DA neurons (36%) and GABA neurons (74%) were co-modulated by more than one behavioral variable *Figure 3—figure supplement 3E* (*Engelhard et al., 2019*).

Overall, the regression analysis confirmed the results of the permutation analysis regarding the significant effect of trajectory on midbrain neuronal activity during memory-dependent decisions. In addition, it demonstrated that the remaining independent variables included in our model cannot fully explain trajectory-specific firing activities in the delay period of the memory task.

