## [Editor Report · eLife assessment]

This study characterized the activity of optogenetically identified dopaminergic and GABAergic neurons in the ventral tegmental area in mice performing a memory-guided T-maze task, and shows that subpopulations of dopaminergic and GABAergic neurons exhibited choice-related activity during the delay period, consistent with some previous studies (e.g. Morris et al., 2006, Parker et al., 2016). The authors demonstrate that these delay-period activities were enhanced when the task requires short-term memory. The results are **convincing** and this study provides **important** results regarding the nature of delay-period activity in the task.

---

## [Referee Report · Reviewer #1 (Public Review)]

Midbrain dopamine neurons have attracted attention as a part of the brain's reward system. A different line of research, on the other hand, has shown that these neurons are also involved in higher cognitive functions such as short-term memory. However, these neurons are thought not to encode short-term memory itself because they just exhibit a phasic response in short-term memory tasks, which cannot seem to maintain information during the memory period. To understand the role of dopamine neurons in short-term memory, the present study investigated the electrophysiological property of these neurons in rodents performing a T-maze version of short-term memory task, in which a visual cue indicated which arm (left or right) of the T-maze was associated with a reward. The animal needed to maintain this information while they were located between the cue presentation position and the selection position of the T-maze. The authors found that the activity of some dopamine neurons changed depending on the information while the animals were located in the memory position. This dopamine neuron modulation was unable to explain the motivation or motor component of the task. The authors concluded that this modulation reflected the information stored as short-term memory.

Comments on revised submission:

The authors adequately responded to all my concerns in the revised manuscript.

---

## [Referee Report · Reviewer #2 (Public Review)]

The authors phototag DA and GABA neurons in the VTA in mice performing a t-maze task, and report choice-specific responses in the delay period of a memory-guided task, more so that in a variant task w/o a memory component. Overall, I found the results convincing. While showing responses that are choice selective in DA neurons is not entirely novel (e.g. Morris et al NN 2006, Parker et al NN 2016), the fact that this feature is stronger when there is a memory requirement is an interesting and a novel observation.

---

## [Author Response]

**Reviewer #1 (Public Review)**
Midbrain dopamine neurons have attracted attention as a part of the brain's reward system. A different line of research, on the other hand, has shown that these neurons are also involved in higher cognitive functions such as short-term memory. However, these neurons are thought not to encode short-term memory itself because they just exhibit a phasic response in short-term memory tasks, which cannot seem to maintain information during the memory period. To understand the role of dopamine neurons in short-term memory, the present study investigated the electrophysiological property of these neurons in rodents performing a T-maze version of a short-term memory task, in which a visual cue indicated which arm (left or right) of the T-maze was associated with a reward. The animal needed to maintain this information while they were located between the cue presentation position and the selection position of the T-maze. The authors found that the activity of some dopamine neurons changed depending on the information while the animals were located in the memory position. This dopamine neuron modulation was unable to explain the motivation or motor component of the task. The authors concluded that this modulation reflected the information stored as short-term memory.I was simply surprised by their finding because these dopamine neurons are similar to neurons in the prefrontal cortex that store memory information with sustained activity. Dopamine neurons are an evolutionally conserved structure, which is seen even in insects, whereas the prefrontal cortex is developed mainly in the primate. I feel that their findings are novel and would attract much attention from readers in the field. But the authors need to conduct additional analyses to consolidate their conclusion.

We thank reviewer #1 for the positive assessment and for the valuable and constructive comments on our manuscript.

**Reviewer #1 (Recommendations to The Authors)**
(1) The authors found the dopamine neuron modulation that reflected the memory information during the delay period. Here the dopamine neuron activity was aligned by the position, not by time, in which the animals needed to maintain the information. Usually, the activity was aligned by time, and many studies found that dopamine neurons exhibited a short duration burst in response to rewards and behaviorally relevant stimuli including visual cues presented in short-term memory tasks. For comparison, I (and probably other readers) want to see the time-aligned dopamine neuron modulation that reflected the memory information. Did the modulation still exist? Did it have a long duration? The authors just showed the time-aligned "population" activity that exhibited no memory-dependent modulation.

We agree that the point raised by the reviewer is important. To address this question, we added a new paragraph to the Methods section titled “Methodological considerations” (in line 793 of the revised manuscript), where we explain the caveats of using time alignment in the T-maze task study. We also created a new sup figure 5 to clarify our argument. As the figure shows, we did not observe major differences in the firing rates when they were arranged by position or time. More importantly, we did not detect brief bursts of activity in response to the visual cue which could reflect an RPE signaling scheme. Our interpretation is that in the T-maze task, DA neurons encode “miniature” RPE signals between successive states in the T-maze, which are hard to detect, especially when neurons receive a continuous sensory input during trials.

(2) Several studies have reported that dopamine neurons at different locations encode distinct signals even within the VTA or SNr. Were the locations of dopamine neurons maintaining the memory information different from those of other dopamine neurons?

We thank the reviewer’s comment. Indeed, there is evidence from recent studies demonstrating that DA neurons form functional and anatomical clusters in the VTA and SN. Following the reviewer’s advice, we report the anatomical structure of memory and non-memory-specific neurons in the revised manuscript. You can read these results in the paragraph “Anatomical organization of trajectory-specific neurons.” in the “Results” section (in line 383 of the revised manuscript) and in the new sup figure 11. We only observed a clear functional-anatomical segregation in GABA neurons, but not in DA neurons. But we should note that the absence of segregation in the DA neurons could be accounted for by the fact that we recorded mostly from the lateral VTA, therefore we do not have any numbers from the medial VTA.

(3a) Did the dopamine neurons maintaining the memory information respond to reward?

We believe that we have already provided the data that can partially answer this question by correlating the firing rate difference between the reward and memory delay sections. This result was described in the “Neuronal activities in delay and reward are unrelated.” paragraph and in Figure 6. Moreover, motivated by the reviewer’s question, we also performed additional analysis, which is included in the revised manuscript. Briefly, we clustered significant responses between the memory delay and reward sections (Category 1: Left-signif, R-signif or No-signif / Category 2: Memory delay or Reward). We discovered that only a very small number of neurons showed the same significant trajectory preference in the memory delay and reward sections (i.e., significant preference for left trials in the memory delay and significant preference for the left reward). In fact, more significant neurons showed a preference for opposite trajectories (i.e. significant preference for left trials in memory delay and a significant preference for right rewards). A description of the new results is included in the “Neuronal activities in delay and reward are unrelated.” paragraph (in line 349 of the revised manuscript) and in the new supplementary Figure 11.

(3b) Did they encode reward prediction error? The relationship between the present data and the conventional theory may be valuable.

We understand that the readers of this study will come up with the question of how memory-specific activities are related to RPE signaling. However, the T-maze task we used in this research was designed for studying working memory and was not adequate to extract information about the RPE signaling of DA neurons.

RPE signaling is mainly studied in Pavlovian conditioning. These are low-dimensional tasks with usually four (4) states (state1: ITI, state2: trial start, state3: stimulus presentation, state4: reward delivery). Evidence of RPE signaling is extracted from the firing activity of states 3 and 4 (which is theorized to be related to the difference in the values for states 3 and 4).

However, in the T-maze task, the number of states is hard to define and practically countless. In these conditions, it has been suggested that numerous small RPEs are signaled while the mice navigate the maze; Thus, they are very difficult to detect. To our knowledge, only Kim et al 2020, Cell, vol183, pg1600, managed to detect the RPE signaling activity of DA neurons while mice were teleported in a virtual corridor.

Another confounding factor in extracting RPE signals in the T-maze task is that the environment is high-dimensional and DA neurons are multitasking. Therefore, it is likely that RPE signaling could be masked by other parallel encoding schemes.

We have added these descriptions in the “Methodological considerations” (in line 793 of the revised manuscript).

(4) Did the dopamine neurons maintaining the memory information (left or right) prefer a contralateral direction like neurons in the motor cortex?

We thank the reviewer for this comment. Indeed, the majority of the memory-specific DA neurons showed a preference for the contralateral direction. We report this result in the legend of the new sup fig 10 (in line 1668 of the revised manuscript).

(5) As shown in Table S2, the proportion of GABA neurons maintaining the memory information (left or right during delay) was much larger than that of dopamine neurons. It seems to be strange because the main output neurons in the VTA are dopaminergic. What is the role of these GABA neurons?

We thank the reviewer for pointing this out. The present study shows that in both populations a sizeable portion of neurons show memory-specific encoding activities. However, the percentage of memory-encoding GABA neurons is more than twice as large as in the DA neurons. Moreover, we show that GABA neurons are functionally and anatomically segregated.

From this evidence, one could raise the hypothesis that the GABA neurons have a primary role and that the activity of DA neurons is a collateral phenomenon, triggered in a sequence of events within the VTA network. To characterize the (1) role and (2) importance of GABA neurons in memory-guided behavior, one should first identify the afferent and efferent projections of these cells in great detail. Unfortunately, we do not provide anatomical evidence.

So far, with the electrophysiological data we have collected (unit and field recordings), we can address an alternative hypothesis. It has been reported earlier (but we have also observed) that the VTA circuit engages in behaviorally related network oscillations which range from 0.4Hz up to 100Hz. Converging evidence from different brain regions, in vitro preparations but also in vivo recordings agree that local networks of inhibitory neurons are crucial for the generation, maintenance, and spectral control of network oscillations. Ongoing analysis, which we hope will lead to a publication, is looking for the behavioral correlates of network oscillations on the T-maze task, as well as the correlation of single-unit firing activity to the field oscillations. We expect to detect a higher field-unit coherence in GABA neurons, which could explain their stronger engagement in memory-specific encoding activity.

The potential role of GABA neurons in network oscillations is discussed in the revised manuscript in a newly added paragraph in line 564.

**Reviewer #2 (Public Review)**
The authors phototag DA and GABA neurons in the VTA in mice performing a t-maze task, and report choice-specific responses in the delay period of a memory-guided task, more so than in a variant task w/o a memory component. Overall, I found the results convincing. While showing responses that are choice selective in DA neurons is not entirely novel (e.g. Morris et al NN 2006, Parker et al NN 2016), the fact that this feature is stronger when there is a memory requirement is an interesting and novel observation.I found the plots in 3B misleading because it looks like the main result is the sequential firing of DA neurons during the Tmaze. However, many of the neurons aren't significant by their permutation test. Often people either only plot the neurons that are significant, or plot with cross-validation (ie sort by half of the trials, and plot the other half).Relatedly, the cross-task comparisons of sequences (Fig, 4,5) are hampered by the fact that they sort in one task, then plot in the other, which will make the sequences look less robust even if they were equally strong. What happens if they swap which task's sequences they use to order the neurons? I do realize they also show statistical comparisons of modulated units across tasks, which is helpful.

We thank reviewer #2 for the valuable and constructive comments on our manuscript. If, as the reviewer commented, the rate differences between left and right trajectories were only the result we want to claim, there may be a way to show only those whose left and right are significant. However, the sequential activity is also one of the points we wanted to display. We did not emphasize this result because it has already been shown by Engelhard et al. 2019. However, after reading the reviewer's comments, we decided to add a few lines in the "Results" (in lines 205 - 215 of the revised manuscript) and "Discussion" (in line 453 of the revised manuscript) describing the sequential activity of the VTA circuit. In those lines, we explained that DA activity is position-specific (resulting in sequential activity) and that a fraction of them also have left-right specificity.

Overall, the introduction was scholarly and did a good job covering a vast literature. But the explanation of t-maze data towards the end of the introduction was confusing. In Line 87, I would not say "in the same task" but "in a similar task" because there are many differences between the tasks in question.

We thank the reviewer for pointing out this mistake. In the revised manuscript, we replaced “in the same task” with “in a similar task” (in line 85 of the revised manuscript).

And not clear what is meant by "by averaging neuronal population activities, none of these computational schemes would have been revealed. " There was trial averaging, at least in Harvey et al. I thought the main result of that paper related to coding schemes was that neural activity was sequential, not persistent. I think it would help the paper to say that clearly.

We admit that this sentence leaves room for misunderstanding. We were mainly referring to DA studies using microdialysis or fiber photometry techniques. We decided to delete this sentence in the revised manuscript.

Also, I'm not aware it was shown that choice selectivity diminishes when the memory demand of the task is removed - please clarify if that is true in both referenced papers.

The reviewer’s remark is correct. None of these reports show explicitly that memory-specific activities are diminished without the memory component. Therefore, we deleted this sentence in the revised manuscript.

If so, an interpretation of this present data could be found in Lee et al biorxiv 2022, which presents a computational model that implies that the heterogeneity in the VTA DA system is a reflection of the heterogeneity found in upstream regions (the state representation), based on the idea that different subsets of DA neurons calculate prediction errors with respect to different subsets of the state representation.

We thank the reviewer for sharing this interpretation. We agree that this theory would support our results. In the revised manuscript we briefly discuss the Lee et al. report (in line 460 of the revised manuscript).

I am surprised only 28% of DA neurons responded to the reward - the reward is not completely certain in this task. This seems lower than other papers in mice (even Pavlovian conditioning, when the reward is entirely certain). It would be helpful if the authors comment on how this number compares to other papers.

In Pavlovian conditioning, neuronal responses to rewards are compared to a relatively quiet period of firing activity (usually the inter-trial interval epoch). As the reviewer pointed out, in the present study, the number of DA neurons responding to reward is smaller compared to the earlier studies. We hypothesize that this is due to our comparison method. We compared the post-reward response to an epoch when the animal was running along the side arms and the majority of neurons were highly active, instead of comparing it to a quiescent baseline epoch.

**Reviewer #2 (Recommendations to The Authors)**
Can you clarify what disparity you are referring to here? "Disparities between this 438 and our study in the proportions of modulated neurons could be attributed to the 439 different recording techniques applied as well as the maze regions of interest; for 440 example, Engelhard et al. analyzed neuronal firing activities in the visual-cue period 441 (Engelhard et al., 2019), whereas we focused on memory delay.". Is it the fact that Engelhard et al did not report choice-selective activity? They did report cue-side-selective activity, with some neurons responsive to cues on one side, and other neurons responsive to cues on the other side. Because there are more cues on the left when the mouse turns left, these neurons do indeed have choice-selective responses.

We thank the reviewer for this comment. We agree that we need to clarify further our argument. As the reviewer pointed out, Engelhard et al identified choice-specific DA neurons. However, they reported the encoding properties of DA neurons only in the visual-cue period and the reward period. Remarkably, although the task has a memory delay, they did not report the neuronal firing activities for this delay period. Instead, in the present study we dedicated most of our analysis to characterizing the firing properties of VTA neurons in the delay period.

Also, in response to your comment, we edited the paragraph where we describe the disparities between our study and Engelhard et al (in line 466 in the revised manuscript).

I don't think this sentence of intro is needed since it doesn't really contain new info: "Therefore, we looked for hints 116 of memory-related encoding activities in single DA and GABA neurons by 117 characterizing their firing preference for opposite behavioral choices.".

We agree with the reviewer. Therefore, we deleted this sentence in the revised manuscript.

I didn't understand this line of discussion: "Our evidence does not question the validity of this computational model, since we do not provide evidence of how the selective preference for one response over the other translates into the release site.".

The gating theory is based on experimental evidence of neuronal firing activities of DA neurons but also takes into consideration (to a lesser degree) the pre- and post-synaptic processes at the DA release sites (inverted U-shape of D1R activity). We thought that the reader may come to the conclusion that we question the validity of the gating theory. But this is not our intention, especially when we do not provide important evidence such as (1) the projection sites of DA and GABA neurons and (2) the sequence of events that take place at the synaptic triads following the DA and GABA release.

After reading your comment we came to the conclusion that this sentence should be omitted because it is not within the scope of this study to question the validity of the gating theory. Instead, we dedicated a few lines of text to explaining which components of the gating theory (“update”, “maintenance & manipulation” and “motor preparation”) could be attributed to the trajectory-specific activities in the memory delay of the T-maze task. (section “Activities of midbrain DA neurons in short-term memory” in line 417 of the revised manuscript).

In 1B, please illustrate when the light pulses are on & off?

Following the reviewer’s instruction, we added colored bars on top of the raster plots in Figure 1B, indicating the light induction conditions.

In legend for 6C, please clarify it's a correlation between the difference in R and L choice activity across the epochs (if my understanding is correct).

The reviewer’s understanding is correct. We took this advice into consideration to further clarify the methods of analysis that led to the plot in Figure 6C (in line 1246 in the revised manuscript).